# Active Bayesian Causal Inference

**Christian Toth**
TU Graz

**Lars Lorch**
ETH Zürich

**Christian Knoll**
TU Graz

**Andreas Krause**
ETH Zürich

**Franz Pernkopf**
TU Graz

**Robert Peharz**[*]
TU Graz

**Julius von Kügelgen**[*]
MPI for Intelligent Systems, Tübingen
University of Cambridge

## Abstract

Causal discovery and causal reasoning are classically treated as separate and con-secutive tasks: one first infers the causal graph, and then uses it to estimate causal effects of interventions. However, such a two-stage approach is uneconomical, especially in terms of actively collected interventional data, since the causal query of interest may not require a fully-specified causal model. From a Bayesian per-spective, it is also unnatural, since a causal query (e.g., the causal graph or some causal effect) can be viewed as a latent quantity subject to posterior inference—other unobserved quantities that are not of direct interest (e.g., the full causal model) ought to be marginalized out in this process and contribute to our epistemic uncertainty. In this work, we propose Active Bayesian Causal Inference (ABCI), a *fully-Bayesian active learning framework for integrated causal discovery and rea-soning*, which jointly infers a posterior over causal models and queries of interest. In our approach to ABCI, we focus on the class of causally-sufficient, nonlinear additive noise models, which we model using Gaussian processes. We sequentially design experiments that are maximally informative about our target causal query, collect the corresponding interventional data, and update our beliefs to choose the next experiment. Through simulations, we demonstrate that our approach is more data-efficient than several baselines that only focus on learning the full causal graph. This allows us to accurately learn downstream causal queries from fewer samples while providing well-calibrated uncertainty estimates for the quantities of interest.

## 1 Introduction

Causal reasoning, that is, answering causal queries such as the effect of a particular intervention, is a fundamental scientific quest [3, 36, 39, 49]. A rigorous treatment of this quest requires a reference causal model, typically consisting at least of (i) a causal diagram, or directed acyclic graph (DAG), capturing the qualitative causal structure between a system's variables [55] and (ii) a joint distribution that is Markovian w.r.t. this causal graph [75]. Other frameworks additionally model (iii) the functional dependence of each variable on its causal parents in the graph [56, 83]. If the graph is not known from domain expertise, causal discovery aims to infer it from data [48, 75]. However, given only passively-collected observational data and no assumptions on the data-generating process, causal discovery is limited to recovering the Markov equivalence class (MEC) of DAGs implying the conditional independences present in the data [75]. Additional assumptions like linearity can render the graph identifiable [37, 61, 71, 86] but are often hard to falsify, thus leading to risk of

---

[*]Shared last author.

Correspondence to: {`christian.toth,robert.peharz`}`@tugraz.at`, `jvk@tue.mpg.de`
Code available at: https://www.github.com/chritoth/active-bayesian-causal-inference

36th Conference on Neural Information Processing Systems (NeurIPS 2022).

misspecification. These shortcomings motivate learning from experimental (interventional) data, which enables recovering the true causal structure [16, 17, 31]. Since obtaining interventional data is costly in practice, we study the active learning setting, in which we sequentially design and perform interventions that are most informative for the target causal query [1, 26, 31, 32, 50, 79].

Classically, causal discovery and reasoning are treated as separate, consecutive tasks that are studied by different communities. Prior work on experimental design has thus focused either purely on causal reasoning—that is, how to best design experimental studies if the causal graph is known?—or purely on causal discovery, whenever the graph is unknown [35, 61]. In the present work, we consider the more general setting in which we are interested in performing causal reasoning but do not have access to a reference causal model a priori. In this case, causal discovery can be seen as a means to an end rather than as the main objective. Focusing on actively learning the *full* causal model to enable subsequent causal reasoning can thus be disadvantageous for two reasons. First, wasting samples on learning the full causal graph is suboptimal if we are only interested in specific aspects of the causal model. Second, causal discovery from small amounts of data entails significant epistemic uncertainty—for example, incurred by low statistical test power or multiple highly-scoring DAGs—which is not taken into account when selecting a single reference causal model [2, 21].

In this work, we propose *Active Bayesian Causal Inference* (ABCI), a fully-Bayesian framework for integrated causal discovery and reasoning with experimental design. The basic approach is to put a Bayesian prior over the causal model class of choice, and to cast the learning problem as Bayesian inference over the model posterior. Given the unobserved causal model, we formalize causal reasoning by introducing the *target causal query*, a function of the causal model that specifies the set of causal quantities we are interested in. The model posterior together with the query function induce a *query posterior*, which represents the result of our Bayesian learning procedure. It can be used, e.g., in downstream decision tasks or to derive a MAP solution or suitable expectation. To learn the query posterior, we follow the Bayesian optimal experimental design approach [10, 42] and sequentially choose admissible interventions on the true causal model that are most informative about our target query w.r.t. our current beliefs. Given the observed data, we then update our beliefs by computing the posterior over causal models and queries and use them to design the next experiment.

Since inference in the general ABCI framework is computationally highly challenging, we instantiate our approach for the class of causally-sufficient, nonlinear additive Gaussian noise models [37], which we model using Gaussian processes (GPs) [22, 82]. To perform efficient posterior inference in the combinatorial space of causal graphs, we use a recently proposed framework for differentiable Bayesian structure learning (DiBS) [45] that employs a continuous latent probabilistic graph representation. To efficiently maximise the information gain in the experiment design loop, we rely on Bayesian optimisation [46, 47, 73]. Overall, we highlight the following contributions:

- We propose ABCI as a flexible Bayesian active learning framework for efficiently inferring arbitrary sets of causal queries, subsuming causal discovery and reasoning as special cases (§ 3).

- We provide a fully Bayesian treatment for the flexible class of nonlinear additive Gaussian noise models by leveraging GPs, continuous graph parametrisations, and Bayesian optimisation (§ 4).

- We demonstrate that our approach scales to relevant problem sizes and compares favourably to baselines in terms of efficiently learning the graph, full SCM, and interventional distributions (§ 5).

## 2   Related Work

Causal discovery and reasoning have been widely studied in machine learning and statistics [27, 35, 61, 81]. Given an already collected set of observations, there is a large body of literature on learning causal structure, both in the form of a point estimate [9, 30, 43, 59, 60, 71, 75] and a Bayesian posterior [2, 4, 12, 14, 21, 33, 45]. Given a known causal graph, previous work studies how to estimate treatment effects or counterfactuals [56, 67, 69]. When interventional data is yet to be collected, existing work primarily focuses on the specific task of structure learning—without its downstream use. The concept of (Bayesian) active causal discovery was first considered in discrete [50, 79] or linear [11, 53] models with closed-form marginal likelihoods and later extended to nonlinear causal mechanisms [78, 80], multi-target interventions [77], and general models by using hypothesis testing [23] or heuristics [68]. Graph theoretic works give insights on the interventions required for partial or full identifiability [15–17, 31, 38, 40, 70, 84].

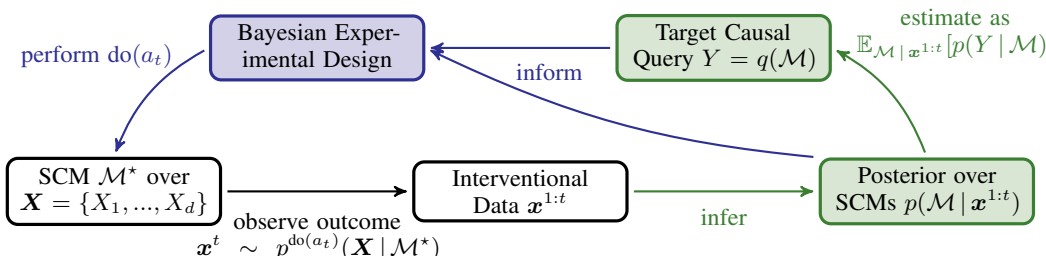

Figure 1: **Overview of the Active Bayesian Causal Inference (ABCI) framework.** At each time step $t$, we use Bayesian experimental design based on our current beliefs to choose a maximally informative intervention $a_t$ to perform. We then collect a finite data sample from the interventional distribution induced by the environment, which we assume to be described by an unknown structural causal model (SCM) $\mathcal{M}^\star$ over a set of observable variables $\boldsymbol{X}$. Given the interventional data $\boldsymbol{x}^{1:t}$ collected from the true SCM $\mathcal{M}^\star$ and a prior distribution over the model class of consideration, we infer the posterior over a target causal query $Y = q(\mathcal{M})$ that can be expressed as a function of the causal model. For example, we may be interested in the graph (causal discovery), the presence of certain edges (partial causal discovery), the full SCM (causal model learning), a collection of interventional distributions or treatment effects (causal reasoning), or any combination thereof.

Beyond learning the complete causal graph, few prior works have studied active causal inference. Concurrent work of Tigas et al. [78] considers experimental design for learning a full SCM parameterised by neural networks. There are significant differences to our approach. In particular, our framework (§ 3) is not limited to the information gain over the full model and provides a fully Bayesian treatment of the functions and their epistemic uncertainty (§ 4). Agrawal et al. [1] consider actively learning a function of the causal graph under budget constraints, though not of the causal mechanisms and only for linear Gaussian models. Conversely, Rubenstein et al. [66] perform experimental design for learning the causal mechanisms after the causal graph has been inferred. Thus, while prior work considers causal discovery and reasoning as separate tasks, ABCI forms an integrated Bayesian approach for learning causal queries through interventions, reducing to previously studied settings in special cases. We further discuss related work in Appx. A.

## 3 Active Bayesian Causal Inference (ABCI) Framework

In this section, we first introduce the ABCI framework in generality and formalize its main concepts and distributional components, which are illustrated in Fig. 1. In § 4, we then describe our particular instantiation of ABCI for the class of causally sufficient nonlinear additive Gaussian noise models.

**Notation.** We use upper-case $X$ and lower-case $x$ to denote random variables and their realizations, respectively. Sets and vectors are written in bold face, $\boldsymbol{X}$ and $\boldsymbol{x}$. We use $p(\cdot)$ to denote different distributions, or densities, which are distinguished by their arguments.

**Causal Model.** To treat causality in a rigorous way, we first need to postulate a mathematically well-defined causal model. Historically hard questions about causality can then be reduced to epistemic questions, that is, what and how much is known about the causal model. A prominent type of causal model is the structural causal model (SCM) [56]. From a Bayesian perspective, an SCM can be viewed as a hierarchical data-generating process involving latent random variables.

**Definition 1** (SCM). An SCM $\mathcal{M}$ over observed endogenous variables $\boldsymbol{X} = \{X_1, \ldots, X_d\}$ and unobserved exogenous variables $\boldsymbol{U} = \{U_1, \ldots, U_d\}$ consists of structural equations, or mechanisms,

$$X_i := f_i(\mathbf{Pa}_i, U_i), \qquad \text{for} \quad i \in \{1, \ldots, d\}, \tag{3.1}$$

which assign the value of each $X_i$ as a deterministic function $f_i$ of its direct causes, or causal parents, $\mathbf{Pa}_i \subseteq \boldsymbol{X} \setminus \{X_i\}$ and $U_i$; and a joint distribution $p(\mathbf{U})$ over the exogenous variables.

Associated with each SCM is a directed causal graph $G$ with vertices $\boldsymbol{X}$ and edges $X_j \to X_i$ if and only if $X_j \in \mathbf{Pa}_i$, which we assume to be acyclic. Any acyclic SCM then induces a unique observational distribution $p(\boldsymbol{X} \mid \mathcal{M})$ over the endogenous variables $\boldsymbol{X}$, which is obtained as the pushforward measure of $p(\boldsymbol{U})$ through the causal mechanisms in Eq. (3.1).

**Interventions.** A crucial aspect of causal models such as SCMs is that they also model the effect of *interventions*—external manipulations to one or more of the causal mechanisms in Eq. (3.1)—which,

in general, are denoted using Pearl's do-operator [56] as $\mathrm{do}(\{X_i = \tilde{f}_i(\mathbf{Pa}_i, U_i)\}_{i \in \mathcal{I}})$ with $\mathcal{I} \subseteq [d]$ and suitably chosen $\tilde{f}_i(\cdot)$. An intervention leads to a new SCM, the so-called interventional SCM, in which the relevant structural equations in Eq. (3.1) have been replaced by the new, manipulated ones. The interventional SCM thus induces a new distribution over the observed variables, the so-called interventional distribution, which is denoted by $p^{\mathrm{do}(a)}(\boldsymbol{X} \,|\, \mathcal{M})$ with $a$ denoting the (set of) intervention(s) $\{X_i = \tilde{f}_i(\mathbf{Pa}_i, U_i)\}_{i \in \mathcal{I}}$. Causal effects, that is, expressions like $\mathbb{E}[X_j | \mathrm{do}(X_i = 3)]$, can then be derived from the corresponding interventional distribution via standard probabilistic inference.

**Being Bayesian with Respect to Causal Models.** The main epistemic challenge for causal reasoning stems from the fact that the true causal model $\mathcal{M}^\star$ is not or not completely known. The canonical response to such epistemic challenges is a Bayesian approach: place a prior $p(\mathcal{M})$ over causal models, collect data $\mathcal{D}$ from the true model $\mathcal{M}^\star$, and compute the posterior via Bayes rule:

$$p(\mathcal{M} \,|\, \mathcal{D}) = \frac{p(\mathcal{D} \,|\, \mathcal{M}) \, p(\mathcal{M})}{p(\mathcal{D})} = \frac{p(\mathcal{D} \,|\, \mathcal{M}) \, p(\mathcal{M})}{\int p(\mathcal{D} \,|\, \mathcal{M}) \, p(\mathcal{M}) \, \mathrm{d}\mathcal{M}}. \tag{3.2}$$

A full Bayesian treatment over $\mathcal{M}$ is computationally delicate, to say the least. We require a way to parameterise the class of models $\mathcal{M}$ while being able to perform posterior inference over this model class. In this paper, we present a fully Bayesian approach for flexibly modelling nonlinear relationships (§ 4).

**Bayesian Causal Inference.** In the causal inference literature, the tasks of causal discovery and causal reasoning are typically considered separate problems. The former aims to learn (parts of) the causal model $\mathcal{M}^\star$, typically the causal graph $G^\star$, while the latter assumes that the relevant parts of $\mathcal{M}^\star$ are already known and aims to identify and estimate some query of interest, typically using only observational data. This separation suggests a two-stage approach of first performing causal discovery and then fixing the model for subsequent causal reasoning. From the perspective of uncertainty quantification and active learning, however, this distinction is unnatural because intermediate, unobserved quantities like the causal model do not contribute to the epistemic uncertainty in the final quantities of interest. Instead, we define a causal query function $q$, which specifies a *target causal query* $Y = q(\mathcal{M})$ as a function of the causal model $\mathcal{M}$. This view thus subsumes and generalises causal discovery and reasoning into a unified framework. For example, possible causal queries are:

*Causal Discovery:* $Y = q_{\mathrm{CD}}(\mathcal{M}) = G$, that is, learning the full causal graph $G$;

*Partial Causal Discovery:* $Y = q_{\mathrm{PCD}}(\mathcal{M}) = \phi(G)$, that is, learning some feature $\phi$ of the graph, such as the presence of a particular (set of) edge(s);

*Causal Model Learning:* $Y = q_{\mathrm{CML}}(\mathcal{M}) = \mathcal{M}$, that is, learning the full SCM $\mathcal{M}$;

*Causal Reasoning:* $Y = q_{\mathrm{CR}}(\mathcal{M}) = \{X_j^{\mathrm{do}(\boldsymbol{X}_{\mathcal{I}(j)} = \boldsymbol{\psi}_j)}\}_{j \in \mathcal{J}}$, that is, learning a set of interventional variables $X_j$ induced by $\mathcal{M}$ under $\mathrm{do}(\boldsymbol{X}_{\mathcal{I}(j)} = \boldsymbol{\psi}_j)$.[2]

Given a causal query, Bayesian inference naturally extends to our learning goal, the *query posterior*:

$$p(Y \,|\, \mathcal{D}) = \int p(Y \,|\, \mathcal{M}) \, p(\mathcal{M} \,|\, \mathcal{D}) \, \mathrm{d}\mathcal{M} = \mathbb{E}_{\mathcal{M} \,|\, \mathcal{D}}[\, p(Y \,|\, \mathcal{M})\,]. \tag{3.3}$$

Evidently, computing Eq. (3.3) constitutes a hard computational problem in general, as we need to marginalise out the causal model. In § 4, we introduce a practical implementation for a restricted causal model class, informed by this challenge.

**Identifiability of causal models and queries.** A crucial concept is that of *identifiability* of a model class, which refers to the ability to uniquely recover the true model in the limit of infinitely many observations from it [25].[3] In the context of our setting, if the class of causal models $\mathcal{M}$ is identifiable, the model posterior $p(\mathcal{M} \,|\, \mathcal{D})$ in Eq. (3.2) and hence, assuming $q(\cdot)$ is deterministic,

---

[2]The return value of $q$ is a set of realisations of the respective random variables. In principle, the set $\mathcal{J}$ can be uncountable, subsuming interventional distributions for a continuous set of intervention values, possibly on different variables. However, instead of having an uncountable set $\mathcal{J}$ for a continuous set of intervention values, it may be more practical to have a finite set $\mathcal{J}$ for intervention targets and to assume a distribution over intervention values $\boldsymbol{\psi}_j \sim p_j(\boldsymbol{\psi})$ as we do in § 4.2 and § 5.

[3]It is worth pointing out that the term "identifiability" is sometimes used differently in the causal inference literature: within causal discovery, it typically refers to *structure identifiability*, that is, recovering only the causal graph; in the context of causal reasoning, on the other hand, it typically refers to whether an interventional (or counterfactual) query can be *expressed in terms of known quantities*, usually involving only the observational distribution. Here, we will use the term in its (original) statistical sense to refer to *identifiability of models*.

also the query posterior $p(Y \mid \mathcal{D})$ in Eq. (3.3) will collapse and converge to a point mass on their respective true values $\mathcal{M}^\star$ and $q(\mathcal{M}^\star)$, given infinite data and provided the true model has non-zero mass under our prior, $p(\mathcal{M}^\star) > 0$. Given only *observational* data, causal models are notoriously unidentifiable in general: without further assumptions on $p(\mathbf{U})$ and the structural form of Eq. (3.1), neither the graph nor the mechanisms can be recovered. In this case, $p(\mathcal{M} \mid \mathcal{D})$ may only converge to an equivalence class of models that cannot be further distinguished. Note, however, that even in this case, $p(Y \mid \mathcal{D})$ may still sometimes collapse, for example, if the Markov equivalence class (MEC) of graphs is identifiable (under causal sufficiency) and our query concerns the presence of a particular edge which is shared by all graphs in the MEC.

**Active Learning with Sequential Interventions.** Rather than collect a large observational dataset, we seek to leverage experimental data, which can help resolve some of the aforementioned identifiability issues and facilitate learning our target causal query more quickly, even if the model is identifiable. Since obtaining experimental data is costly in practice, we study the active learning setting in which we sequentially design experiments in the form of interventions $a_t$.[4] At each time step $t$, the outcome of this experiment $a_t$ is a batch $\boldsymbol{x}^t$ of $N_t$ i.i.d. observations from the true interventional distribution:

$$\boldsymbol{x}^t = \{\boldsymbol{x}^{t,n}\}_{n=1}^{N_t}, \qquad \boldsymbol{x}^{t,n} \overset{\text{i.i.d.}}{\sim} p^{\text{do}(a_t)}(\boldsymbol{X} \mid \mathcal{M}^\star) \qquad (3.4)$$

Crucially, we design the experiment $a_t$ to be *maximally informative* about our target causal query $Y$. In our Bayesian setting, this is naturally formulated as maximising the myopic information gain from the next intervention, that is, the mutual information between $Y$ and the outcome $\boldsymbol{X}^t$ [10, 42]:

$$\max_{a_t} \mathrm{I}(Y; \boldsymbol{X}^t \mid \boldsymbol{x}^{1:t-1}) \qquad (3.5)$$

where $\boldsymbol{X}^t$ follows the predictive interventional distribution of the Bayesian causal model ensemble at time $t-1$ under intervention $a_t$, which is given by

$$\boldsymbol{X}^t \sim p^{\text{do}(a_t)}(\boldsymbol{X} \mid \boldsymbol{x}^{1:t-1}) \propto \int p^{\text{do}(a_t)}(\boldsymbol{X} \mid \mathcal{M}) \, p(\mathcal{M} \mid \boldsymbol{x}^{1:t-1}) \, \mathrm{d}\mathcal{M}. \qquad (3.6)$$

By maximising Eq. (3.5), we collect experimental data and infer our target causal query $Y$ in a highly efficient, goal-directed manner.

# 4 Tractable ABCI for Nonlinear Additive Noise Models

Having described the general ABCI framework and its conceptual components, we now detail how to instantiate ABCI for a flexible model class that still allows for tractable, approximate inference. This requires us to specify (i) the class of causal models we consider in Eq. (3.1), (ii) the types of interventions $a_t$ we consider at each step and the corresponding interventional likelihood in Eq. (3.4), (iii) our prior distribution $p(\mathcal{M})$ over models, (iv) how to perform tractable inference of the model posterior in Eq. (3.2), and finally (v) how to maximise the information gain in Eq. (3.5) for experimental design.

**Model Class and Parametrisation.** In the following, we consider nonlinear additive Gaussian noise models [37] of the form

$$X_i := f_i(\mathbf{Pa}_i) + U_i, \qquad \text{with} \qquad U_i \sim \mathcal{N}(0, \sigma_i^2) \qquad \text{for} \quad i \in \{1, \dots, d\}, \qquad (4.1)$$

where the $f_i$'s are smooth, nonlinear functions and the $U_i$'s are assumed to be mutually independent. The latter corresponds to the assumption of causal sufficiency, or no hidden confounding. Any model $\mathcal{M}$ in this model class can be parametrised as a triple $\mathcal{M} = (G, \boldsymbol{f}, \boldsymbol{\sigma}^2)$, where $G$ is a causal DAG, $\boldsymbol{f} = (f_1, \dots, f_d)$ is a vector of functions defined over the parent sets implied by $G$, and $\boldsymbol{\sigma}^2 = (\sigma_1^2, \dots, \sigma_d^2)$ contains the Gaussian noise variances. Provided that the $f_i$ are nonlinear and not constant in any of their arguments, the model is identifiable almost surely [37, 62].

**Interventional Likelihood.** We support the realistic setting where only a subset $\boldsymbol{W} \subseteq \boldsymbol{X}$ of all variables are actionable, that is, can be intervened upon.[5] We consider hard interventions of the form $\mathrm{do}(a_t) = \mathrm{do}(\boldsymbol{X}_{\mathcal{I}} = \boldsymbol{x}_{\mathcal{I}})$ that fix a subset $\boldsymbol{X}_{\mathcal{I}} \subseteq \boldsymbol{W}$ to a constant $\boldsymbol{x}_{\mathcal{I}}$. Due to causal sufficiency, the interventional likelihood under such hard interventions $a_t$ factorises over the causal graph $G$ and is given by the g-formula [64] or truncated factorisation [75]:

$$p^{\text{do}(a_t)}(\boldsymbol{X} \mid G, \boldsymbol{f}, \boldsymbol{\sigma}^2) = \mathbb{I}\{\boldsymbol{X}_{\mathcal{I}} = \boldsymbol{x}_{\mathcal{I}}\} \prod_{j \notin \mathcal{I}} p(X_j \mid f_j(\mathbf{Pa}_j^G), \sigma_j^2). \qquad (4.2)$$

---

[4]Note that restricting to $a_t = \varnothing$ amounts to learning from observational data as a special case.

[5]In principle, the set of actionable variables might even change over time, in which case they are denoted $\boldsymbol{W}_t$.

The last term in Eq. (4.2) is given by $\mathcal{N}(X_j \mid f_j(\mathbf{Pa}_j^G), \sigma_j^2)$, due to the Gaussian noise assumption. Let $\boldsymbol{x}^{1:t}$ be the entire dataset, collected up to time $t$. The likelihood of $\boldsymbol{x}^{1:t}$ is then given by

$$p(\boldsymbol{x}^{1:t} \mid G, \boldsymbol{f}, \boldsymbol{\sigma}^2) = \prod_{\tau=1}^{t} p^{\mathrm{do}(a_\tau)}(\boldsymbol{x}^\tau \mid G, \boldsymbol{f}, \boldsymbol{\sigma}^2) = \prod_{\tau=1}^{t} \prod_{n=1}^{N_t} p^{\mathrm{do}(a_\tau)}(\boldsymbol{x}^{\tau,n} \mid G, \boldsymbol{f}, \boldsymbol{\sigma}^2). \qquad (4.3)$$

**Structured Model Prior.** To specify our prior, we distinguish between root nodes $X_i$, for which $\mathbf{Pa}_i = \varnothing$ and thus $f_i = \mathrm{const}$, and non-root nodes $X_j$. For a given causal graph $G$, we denote the index set of root nodes by $\mathbf{R}(G) = \{i \in [d] : \mathbf{Pa}_i^G = \varnothing\}$ and that of non-root nodes by $\mathbf{NR}(G) = [d] \setminus \mathbf{R}(G)$. We then place the following structured prior over SCMs $\mathcal{M} = (G, \boldsymbol{f}, \boldsymbol{\sigma}^2)$:

$$p(\mathcal{M}) = p(G) \, p(\boldsymbol{f}, \boldsymbol{\sigma}^2 \mid G) = p(G) \prod_{i \in \mathbf{R}(G)} p(f_i, \sigma_i^2 \mid G) \prod_{j \in \mathbf{NR}(G)} p(f_j \mid G) p(\sigma_j^2 \mid G). \qquad (4.4)$$

Here, $p(G)$ is a prior over graphs and $p(\boldsymbol{f}, \boldsymbol{\sigma}^2 \mid G)$ is a prior over the functions and noise variances. We factorise our prior conditional on $G$ as in Eq. (4.4) not only to allow for a separate treatment of root vs. non-root nodes, but also to share priors across similar graphs. Whenever $\mathbf{Pa}_i^{G_1} = \mathbf{Pa}_i^{G_2}$, we set $p(f_i, \sigma_i^2 \mid G_1) = p(f_i, \sigma_i^2 \mid G_2)$. As a consequence, the posteriors are also shared, which substantially reduces the computational cost in practice (see Appx. E.2 for details). Our prior also encodes the beliefs that $\{f_i, \sigma_i^2\} \perp\!\!\!\perp \{f_{i'}, \sigma_{i'}^2\} \mid G$ for $i \neq i' \in [d]$ and that $f_j \perp\!\!\!\perp \sigma_j^2 \mid G$ for $j \in \mathbf{NR}(G)$ which is motivated by the principle of independent causal mechanisms [61] and the causal sufficiency assumption. Our specific choices for the different factors on the RHS of Eq. (4.4) are guided by ensuring tractable inference and described in more detail below.

**Model Posterior.** Given collected data $\boldsymbol{x}^{1:t}$, we can update our beliefs and quantify our uncertainty in $\mathcal{M}^\star$ by inferring the posterior $p(\mathcal{M} \mid \boldsymbol{x}^{1:t})$ over SCMs $\mathcal{M} = (G, \boldsymbol{f}, \boldsymbol{\sigma}^2)$, which can be written as[6]

$$p(\mathcal{M} \mid \boldsymbol{x}^{1:t}) = p(G \mid \boldsymbol{x}^{1:t}) \prod_{i \in \mathbf{R}(G)} p(f_i, \sigma_i^2 \mid \boldsymbol{x}^{1:t}, G) \prod_{j \in \mathbf{NR}(G)} p(f_j, \sigma_j^2 \mid \boldsymbol{x}^{1:t}, G). \qquad (4.5)$$

For root nodes $i \in \mathbf{R}(G)$, posterior inference given the graph is straightforward. We have $f_i = \mathrm{const}$, so $f_i$ can be viewed as the mean of $U_i$. We thus place conjugate normal-inverse-gamma $\mathrm{N}\text{-}\Gamma^{-1}(\mu_i, \lambda_i, \alpha_i^{\mathrm{R}}, \beta_i^{\mathrm{R}})$ priors on $p(f_i, \sigma_i^2 \mid G)$, which allows us to analytically compute the root node posteriors $p(f_i, \sigma_i^2 \mid \boldsymbol{x}^{1:t}, G)$ in Eq. (4.5) given the hyperparameters $(\boldsymbol{\mu}, \boldsymbol{\lambda}, \boldsymbol{\alpha}^{\mathrm{R}}, \boldsymbol{\beta}^{\mathrm{R}})$ [51].

The posteriors over graphs and non-root nodes $j \in \mathbf{NR}(G)$ are given by

$$p(G \mid \boldsymbol{x}^{1:t}) = \frac{p(\boldsymbol{x}^{1:t} \mid G) \, p(G)}{p(\boldsymbol{x}^{1:t})}, \qquad p(f_j, \sigma_j^2 \mid \boldsymbol{x}^{1:t}, G) = \frac{p(\boldsymbol{x}^{1:t} \mid G, f_j, \sigma_j^2) \, p(f_j, \sigma_j^2 \mid G)}{p(\boldsymbol{x}^{1:t} \mid G)}. \qquad (4.6)$$

Computing these posteriors is more involved and discussed in the following.

## 4.1 Addressing Challenges for Posterior Inference with GPs and DiBS

The posterior distributions in Eq. (4.6) are intractable to compute in general due to the marginal likelihood and evidence terms $p(\boldsymbol{x}^{1:t} \mid G)$ and $p(\boldsymbol{x}^{1:t})$, respectively. In the following, we will address these challenges by means of appropriate prior choices and approximations.

**Challenge 1: Marginalising out the Functions.** The marginal likelihood $p(\boldsymbol{x}^{1:t} \mid G)$ reads

$$p(\boldsymbol{x}^{1:t} \mid G) = \int p(\boldsymbol{x}^{1:t} \mid G, f_j, \sigma_j^2) \, p(f_j \mid G) \, p(\sigma_j^2 \mid G) \, \mathrm{d}f_j \, \mathrm{d}\sigma_j^2 \qquad (4.7)$$

and requires evaluating integrals over the function domain. We use Gaussian processes (GPs) [82] as an elegant way to solve this problem, as GPs flexibly model *nonlinear* functions while offering convenient analytical properties. Specifically, we place a $\mathcal{GP}(0, k_j^G(\cdot, \cdot))$ prior on $p(f_j \mid G)$, where $k_j^G(\cdot, \cdot)$ is a covariance function over the parents of $X_j$ with kernel parameters $\boldsymbol{\kappa}_j$. As is common, we refer to $(\boldsymbol{\kappa}_j, \sigma_j^2)$ as the GP-hyperparameters. In addition, we place $\mathrm{Gamma}(\alpha_j^\sigma, \beta_j^\sigma)$ and $\mathrm{Gamma}(\boldsymbol{\alpha}_j^\kappa, \boldsymbol{\beta}_j^\kappa)$ priors on $p(\sigma_i^2 \mid G)$ and $p(\boldsymbol{\kappa}_i \mid G)$ and collect their parameters in $(\boldsymbol{\alpha}^{\mathrm{GP}}, \boldsymbol{\beta}^{\mathrm{GP}})$.

---

[6]To avoid further complicating the notation, we write all posteriors and likelihoods in terms of the full data $\boldsymbol{x}^{1:t}$. However, only observations of $X_i$ and $X_j \mid \mathbf{Pa}_j^G$ matter for $i \in \mathbf{R}(G)$ and $j \in \mathbf{NR}(G)$.

The graphical model underlying all variables and hyper-parameters is shown in Fig. 2. For our model class, GPs provide closed-form expressions for the GP-marginal likelihood $p(\boldsymbol{x}^{1:t} \,|\, G, \sigma_j^2, \boldsymbol{\kappa}_j)$, as well as for the GP posteriors $p(f_j \,|\, \boldsymbol{x}^{1:t}, G, \sigma_j^2, \boldsymbol{\kappa}_j)$ and the predictive posteriors over observations $p(\boldsymbol{X} \,|\, \boldsymbol{x}^{1:t}, G, \boldsymbol{\sigma}^2, \boldsymbol{\kappa})$ [82], see Appx. B for details.

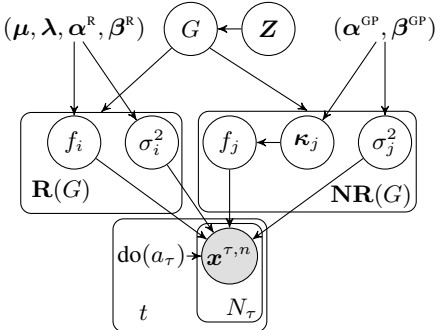

Figure 2: Graphical model of GP-DiBS-ABCI.

**Challenge 2: Marginalising out the GP-Hyperparameters.** While GPs allow for exact posterior inference conditional on a fixed instance of $(\sigma_j^2, \boldsymbol{\kappa}_j)$, evaluating expressions such as $p(f_j \,|\, \boldsymbol{x}^{1:t}, G)$ requires marginalising out these GP-hyperparameters from the GP-posterior. In general, this is intractable to do exactly, as there is no analytical expression for $p(\sigma_j^2, \boldsymbol{\kappa}_j \,|\, \boldsymbol{x}^{1:t}, G)$. To tackle this, we approximate such terms using a maximum a posteriori (MAP) point estimate $(\hat{\sigma}_j^2, \hat{\boldsymbol{\kappa}}_j)$ obtained by performing gradient ascent on the unnormalised log posterior

$$\nabla \log p(\sigma_j^2, \boldsymbol{\kappa}_j \,|\, \boldsymbol{x}^{1:t}, G) = \nabla \log p(\boldsymbol{x}^{1:t} \,|\, G, \sigma_j^2, \boldsymbol{\kappa}_j) + \nabla \log p(\sigma_j^2, \boldsymbol{\kappa}_j \,|\, G) \tag{4.8}$$

according to a predefined update schedule, see Alg. 1. More specifically,

$$p(f_j \,|\, \boldsymbol{x}^{1:t}, G) = \int p(f_j \,|\, \boldsymbol{x}^{1:t}, G, \sigma_j^2, \boldsymbol{\kappa}_j) p(\sigma_j^2, \kappa_j \,|\, \boldsymbol{x}^{1:t}, G) \, \mathrm{d}\sigma_j^2 \, \mathrm{d}\boldsymbol{\kappa}_j \approx p(f_j \,|\, \boldsymbol{x}^{1:t}, G, \hat{\sigma}_j^2, \hat{\boldsymbol{\kappa}}_j)$$

**Challenge 3: Marginalising out the Causal Graph.** The evidence $p(\boldsymbol{x}^{1:t})$ is given by

$$p(\boldsymbol{x}^{1:t}) = \sum_G p(\boldsymbol{x}^{1:t} \,|\, G) \, p(G) \tag{4.9}$$

and involves a summation over all possible DAGs $G$. This becomes intractable for $d \geq 5$ variables as the number of DAGs grows super-exponentially in the number of variables [65]. To address this challenge, we employ the recently proposed DiBS framework [45]. By introducing a continuous prior $p(\boldsymbol{Z})$ that models $G$ via $p(G \,|\, \boldsymbol{Z})$ and simultaneously enforces acyclicity of $G$, Lorch et al. [45] show that we can efficiently infer the discrete posterior $p(G \,|\, \boldsymbol{x}^{1:t})$ via $p(\boldsymbol{Z} \,|\, \boldsymbol{x}^{1:t})$ as

$$\mathbb{E}_{G \,|\, \boldsymbol{x}^{1:t}} \left[ \phi(G) \right] = \mathbb{E}_{\boldsymbol{Z} \,|\, \boldsymbol{x}^{1:t}} \left[ \frac{\mathbb{E}_{G \,|\, \boldsymbol{Z}} \left[ p(\boldsymbol{x}^{1:t} \,|\, G) \, \phi(G) \right]}{\mathbb{E}_{G \,|\, \boldsymbol{Z}} \left[ p(\boldsymbol{x}^{1:t} \,|\, G) \right]} \right] \tag{4.10}$$

where $\phi$ is some function of the graph. Since $p(\boldsymbol{Z} \,|\, \boldsymbol{x}^{1:t})$ is a continuous density with tractable gradient estimators, we can leverage efficient variational inference methods such as Stein Variational Gradient Descent (SVGD) for approximate inference [44]. Additional details on DiBS are given in Appx. D.

### 4.2 Approximate Bayesian Experimental Design with Bayesian Optimisation

Following § 3, our goal is to perform experiments $a_t$ that are maximally informative about our target query $Y = q(\mathcal{M})$ by maximising the information gain from Eq. (3.5) given our hitherto collected data $\mathcal{D} := \boldsymbol{x}^{1:t-1}$. In Appx. C, we show that this is equivalent to maximising the following utility function:

$$U(a) = H(\boldsymbol{X}^t \,|\, \mathcal{D}) + \mathbb{E}_{\mathcal{M} \,|\, \mathcal{D}} \left[ \mathbb{E}_{\boldsymbol{X}^t, Y \,|\, \mathcal{M}} \left[ \log \mathbb{E}_{\mathcal{M}' \,|\, \mathcal{D}} \left[ p(\boldsymbol{X}^t, Y \,|\, \mathcal{M}') \right] \right] \right], \tag{4.11}$$

where

$$H(\boldsymbol{X}^t \,|\, \mathcal{D}) = \mathbb{E}_{\mathcal{M} \,|\, \mathcal{D}} \left[ \mathbb{E}_{\boldsymbol{X}^t \,|\, \mathcal{M}} \left[ \log \mathbb{E}_{\mathcal{M}' \,|\, \mathcal{D}} \left[ p(\boldsymbol{X}^t \,|\, \mathcal{M}') \right] \right] \right]$$

denotes the differential entropy of the experiment outcome which depends on $a$ and is distributed as in Eq. (3.6). This surrogate objective can be estimated using a nested Monte Carlo estimator as long as we can sample from and compute $p(Y \,|\, \mathcal{M})$, or alternatively, $p(Y \,|\, \boldsymbol{X}^t, G, \mathcal{D})$. Refer to Appx. C for further details. For example, for $q_{\text{CR}}(\mathcal{M}) = X_j^{\text{do}(X_i = \psi)}$ with $\psi \sim p(\psi)$ a distribution over intervention values, we obtain:

$$\begin{aligned} U_{\text{CR}}(a) = \mathbb{E}_{G \,|\, \mathcal{D}} \big[ \mathbb{E}_{\boldsymbol{X}^t \,|\, G, \mathcal{D}} \big[ &- \log \mathbb{E}_{G' \,|\, \mathcal{D}} \big[ p(\boldsymbol{X}^t \,|\, \mathcal{D}, G') \big] \\ &+ \mathbb{E}_\psi \big[ \mathbb{E}_{X_j \,|\, \boldsymbol{X}^t, G, \mathcal{D}}^{\text{do}(X_i = \psi)} \big[ \log \mathbb{E}_{G' \,|\, \mathcal{D}} \big[ p(\boldsymbol{X}^t \,|\, \mathcal{D}, G) \, p^{\text{do}(X_i = \psi)}(X_j \,|\, \boldsymbol{X}^t, G, \mathcal{D}) \big] \big] \big] \big] \big]. \end{aligned} \tag{4.12}$$

**Algorithm 1:** GP-DiBS-ABCI for nonlinear additive Gaussian noise models

---

**Input:** # of experiments $T$, batch sizes $\{N_t\}_{t=1}^T$, # of latent particles $M$, # of MC samples $K$,
  particle resampling schedule $\{r_t\}_{t=1}^T$, hyperparameter update schedule $\{s_t\}_{t=1}^T$
**Output:** Posterior over target causal query $p(Y \,|\, \boldsymbol{x}^{1:T})$

---

$\boldsymbol{z}^0 \sim p(\boldsymbol{Z})$            ▷`sample initial particles; Eq.` (D.12)
**for** $t = 1$ **to** $T$ **do**
    $a_t \leftarrow \arg\max_{a=(\mathcal{I}, \boldsymbol{x}_{\mathcal{I}})} U(a, \boldsymbol{x}^{1:t-1})$      ▷`design experiment; Eq.` (4.11)
    $\boldsymbol{x}^t \leftarrow \{\boldsymbol{x}^{(t,n)} \sim p^{\mathrm{do}(a_t)}(\boldsymbol{X} \,|\, \mathcal{M}^\star)\}_{n=1}^{N_t}$      ▷`perform experiment; Eq.` (3.4)
    $\boldsymbol{z}^t \leftarrow \boldsymbol{z}^{t-1}$
    **if** $r_t$ **then**
      $\boldsymbol{z}^t \leftarrow$ `resample_particles` $(\boldsymbol{z}^t)$         ▷`see Appx.`E
    **end**
    **repeat**
      $\boldsymbol{G} \leftarrow \{G^{(k,m)} \sim p(G \,|\, \boldsymbol{z}_m^t)\}_{k=1\ m=1}^{K\ \ M}$     ▷`sample graphs; Eq.` (D.11)
      $\boldsymbol{\kappa}, \boldsymbol{\sigma}^2 \leftarrow$ `estimate_hyperparameters`$(\boldsymbol{x}^{1:s_t}, \boldsymbol{G})$    ▷`see Eq.` (4.8)
      $\boldsymbol{z}^t \leftarrow$ `svgd_step`$(\boldsymbol{z}^t, \boldsymbol{x}^{1:t}, \boldsymbol{G}, \boldsymbol{\kappa}, \boldsymbol{\sigma}^2)$     ▷`update latent particles`
    **until** `svgd_convergence`        ▷$\boldsymbol{z}^t$ `now approximate` $p(\boldsymbol{Z} \,|\, \boldsymbol{x}^{1:t})$
**end**

---

Importantly, for specific instances of the query function $q(\cdot)$ discussed in § 3, we can derive simpler utility functions than Eq. (4.11). For example, for $q_{\mathrm{CD}}(\mathcal{M}) = G$ and $q_{\mathrm{CML}}(\mathcal{M}) = \mathcal{M}$, we arrive at

$$U_{\mathrm{CD}}(a) = \mathbb{E}_{G \,|\, \mathcal{D}}\left[\mathbb{E}_{\boldsymbol{X}^t \,|\, G, \mathcal{D}}\left[\log p(\boldsymbol{X}^t \,|\, \mathcal{D}, G) - \log \mathbb{E}_{G' \,|\, \mathcal{D}}\left[p(\boldsymbol{X}^t \,|\, \mathcal{D}, G')\right]\right]\right], \quad (4.13)$$

$$U_{\mathrm{CML}}(a) = \mathbb{E}_{\mathcal{M} \,|\, \mathcal{D}}\left[\mathbb{E}_{\boldsymbol{X}^t \,|\, \mathcal{M}}\left[\log p(\boldsymbol{X}^t \,|\, \mathcal{M}) - \log \mathbb{E}_{G' \,|\, \mathcal{D}}\left[p(\boldsymbol{X}^t \,|\, \mathcal{D}, G')\right]\right]\right], \quad (4.14)$$

where the entropy $\mathbb{E}_{\boldsymbol{X}^t \,|\, \mathcal{M}}[\log p(\boldsymbol{X}^t \,|\, \mathcal{M})]$ can again be efficiently computed given our modelling choices. For brevity, we defer derivations and estimation details to Appxs. C and D.

Finding the optimal experiment $a_t^* = (\mathcal{I}^*, \boldsymbol{x}_{\mathcal{I}}^*)$ requires jointly optimising the utility function corresponding to our query with respect to (i) the set of intervention *targets* $\mathcal{I}$ and (ii) the corresponding intervention *values* $\boldsymbol{x}_{\mathcal{I}}$. This lends itself naturally to a nested, bi-level optimisation scheme [80]:

$$\mathcal{I}^* \in \arg\max_{\mathcal{I}} U(\mathcal{I}, \boldsymbol{x}_{\mathcal{I}}^*), \quad \text{where} \quad \forall \mathcal{I}: \quad \boldsymbol{x}_{\mathcal{I}}^* \in \arg\max_{\boldsymbol{x}_{\mathcal{I}}} U(\mathcal{I}, \boldsymbol{x}_{\mathcal{I}}), \quad (4.15)$$

In the above, we first estimate the optimal intervention values for all candidate intervention targets $\mathcal{I}$ and then select the intervention target that yields the highest utility. The intervention target $\mathcal{I}$ may contain multiple variables, which would yield a combinatorial problem. For simplicity, we consider only single-node interventions, $|\mathcal{I}| = 1$. To find $\boldsymbol{x}_{\mathcal{I}}^*$, we employ Bayesian optimisation [46, 47, 73] to efficiently estimate the most informative intervention value $\boldsymbol{x}_{\mathcal{I}}^*$, see Appx. D.

## 5 Experiments

**Setup.** We evaluate ABCI by inferring the query posterior on synthetic ground-truth SCMs using several different experiment selection strategies. Specifically, we design experiments w.r.t. $U_{\mathrm{CD}}$ (causal discovery), $U_{\mathrm{CML}}$ (causal model learning), and $U_{\mathrm{CR}}$ (causal reasoning); see § 4.2. We compare against baselines which (i) only sample from the observational distribution (OBS) or (ii) pick an intervention target $j$ uniformly at random from $[d] \cup \{\varnothing\}$ and set $X_j = 0$ (RAND FIXED, a weak random baseline used in prior work) or draw $X_j \sim \mathcal{U}(-7, 7)$ (RAND) if $X_j \neq \varnothing$. All methods follow our Bayesian GP-DiBS-ABCI approach from § 4. We sample ground truth SCMs over random scale-free graphs [6] of size $d = 20$, with mechanisms and noise variances drawn from our model prior in Eq. (4.4). In Appx. G, we report additional results for both scale-free and Erdős Renyi random graphs over $d = 10$ resp. $d = 20$ variables. For specific prior choices and simulation details, see Appx. D.

**Metrics.** As ABCI infers a posterior over the target query $Y$, a natural evaluation metric is the Kullback-Leibler divergence (KLD) between the true query distribution and the inferred query posterior, $\mathrm{KL}(p(Y \,|\, \mathcal{M}^\star) \,||\, p(Y \,|\, \boldsymbol{x}^{1:t}))$. We report **Query KLD**, a KLD estimate for target interventional distributions ($q_{\mathrm{CR}}$). As a proxy for the KLD of the SCM posterior ($q_{\mathrm{CML}}$),[7] we report

---

[7]The SCM KLD is either zero, if the SCM posterior collapses onto the true SCM, or infinite, otherwise.

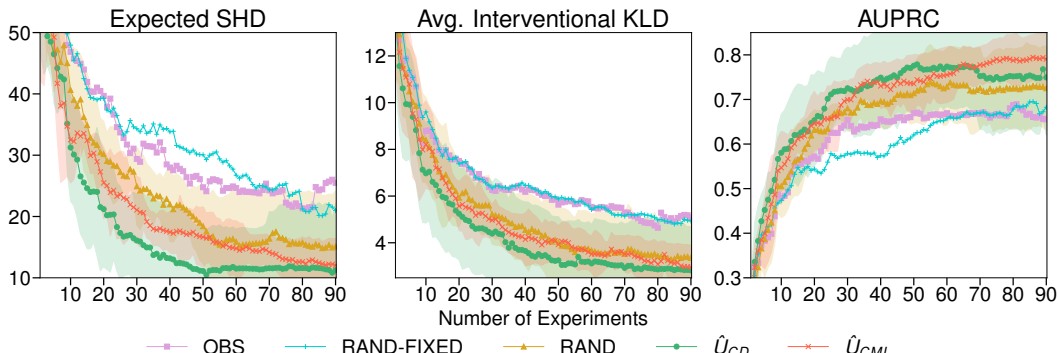

**Figure 3: Causal Discovery and SCM Learning.** Comparison of experimental design strategies for causal discovery ($U_{\mathrm{CD}}$) and causal model learning ($U_{\mathrm{CML}}$) with random and observational baselines on simulated ground truth models with 20 nodes. We initialise all methods with 50 observational samples, and then perform experiments with a batch size of $N_t = 5$. Lines and shaded areas show means and 95% confidence intervals (CIs) across 15 runs (5 randomly sampled ground-truth SCMs with 3 restarts per SCM). CIs for OBS and RAND FIXED baselines are not shown to aid readability; see Fig. 6 in Appx. G for the full figure. **(a) ESHD.** Both our objectives significantly outperform the observational and random baselines. **(b) Average I-KLD.** $U_{\mathrm{CD}}$ significantly outperforms the baselines, whereas $U_{\mathrm{CML}}$ performs only marginally better than RAND. **(c) AUPRC.** Both our strategies perform consistently better than the uninformed selection strategies.

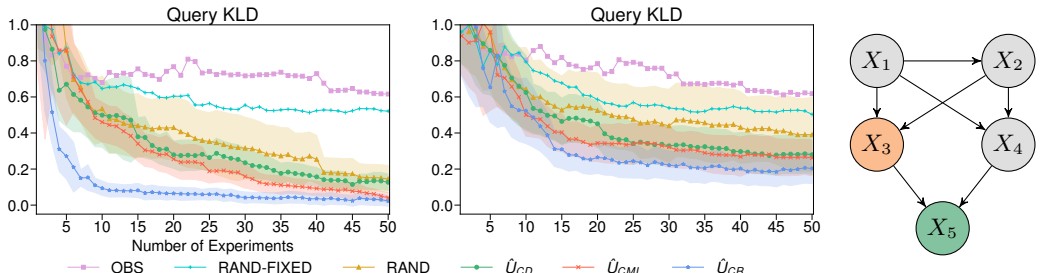

**Figure 4: Learning Interventional Distributions.** (left) Comparison of different methods w.r.t. learning a set of interventional variables $X_5^{\mathrm{do}(X_3=\psi)}$ with $\psi \sim \mathcal{U}[2,5]$ on simulated ground truth models with fixed causal graph (right). We initialise all methods with 5 observational samples, and then perform experiments with a batch size of $N_t = 3$. Lines and shaded areas show means and 95% confidence intervals (CIs) across 30 runs (10 randomly sampled ground truth SCMs with 3 restarts each). CIs for OBS and RAND FIXED baselines are not shown to aid readability; see Figs. 9 and 10 in Appx. G for the full figure. **(a) All nodes actionable.** $U_{\mathrm{CR}}$ significantly outperforms all other methods as expected. $U_{\mathrm{CML}}$ performs second best which, in conjunction with the results from Fig. 3, suggests that $U_{\mathrm{CML}}$ yields a solid base model for performing downstream causal inference tasks. **(b) $X_3$ not actionable.** In this setting, where we cannot directly intervene on the treatment variable of interest, $U_{\mathrm{CR}}$ clearly outperforms all other methods for $\geq 10$ experiments.

the average KLD across all single node interventional distributions $\{p^{\mathrm{do}(X_i=\psi)}(\boldsymbol{X})\}_{i=1}^{d}$, with $\psi \sim \mathcal{U}(-7,7)$ (**Average I-KLD**). We also report the expected structural Hamming distance [13], $\mathbf{ESHD} = \mathbb{E}_{G \mid \boldsymbol{x}^{1:t}}[\mathrm{SHD}(G, G^\star)]$, a commonly used causal discovery metric, and the *area under the precision recall curve* (**AUPRC**). See Appx. F for further details.

**Causal Discovery and SCM Learning (Fig. 3).** In our first experiment, we find that all ABCI-based methods are able to meaningfully learn from small amounts of data, which validates our Bayesian approach. Moreover, performing targeted interventions using experimental design indeed improves performance compared to uninformed experimentation (OBS, RAND FIXED, RAND). Notably, the stronger random baseline (RAND), which also randomises over intervention values, performs well in the considered setting. As expected by the theoretical grounding of the information gain utilities, $U_{\mathrm{CD}}$ identifies the true graph the fastest (as measured by ESHD), whereas $U_{\mathrm{CML}}$ exhibits good scores across all metrics. Further details are given in the caption of Fig. 3.

**Learning Interventional Distributions (Fig. 4).** In our second experiment, we investigate ABCI's causal reasoning capabilities by randomly sampling ground-truth SCMs as described above over the fixed graph shown in Fig. 4 (right), which is *not* known to the methods. Our target query is the set of interventional random variables, or "distributional treatment effects", $X_5^{\mathrm{do}(X_3=\psi)}$ for treatments $\psi \sim \mathcal{U}[2,5]$. The results show that our informed experiment selection strategies significantly outperform the baselines at causal reasoning as measured by the Query KLD. In accordance with the results from Fig. 3 and considering that, once we know the true SCM, we can compute any causal quantity of interest, $U_{\mathrm{CML}}$ seems to provide a reasonable experimental strategy in case the causal query of interest is *not* known a priori. However, our results indicate that if we *do* know our query of interest, then $U_{\mathrm{CR}}$ provides a more efficient experiment design strategy for its estimation, even when the treatment variable of interest is not directly intervenable. In this case, the task is indeed more difficult, as highlighted by the larger Query KLD values across all considered methods.

## 6 Discussion

**Assumptions, Limitations, and Extensions.** In § 4, we have made several assumptions to facilitate tractable inference and showcase the ABCI framework in a relatively simple data-generating process. In particular, our assumptions exclude heteroscedastic noise, unobserved confounding, and cyclic relationships. On the experimental design side, we only considered hard interventions, but for some applications soft interventions [18] are more plausible. On the query side, we only considered interventional distributions. However, SCMs also naturally lend themselves to counterfactual reasoning, so one could also consider counterfactual queries such as the effect of the treatment on the treated [34, 72]. In principle, the ABCI framework as presented in § 3 extends directly to such generalisations. In practice, however, these can be non-trivial to implement, especially with regard to model parametrisation and tractable inference. Since actively performed interventions allow for causal learning even under causal sufficiency violations, we consider this a promising avenue for future work and believe the ABCI framework to be particularly well-suited for exploring it.

**Reflections on the ABCI Framework.** The main conceptual advantages of the ABCI framework are that it is *flexible* and *principled*. By considering general target causal queries, we can precisely specify what aspects of the causal model we are interested in. This conceptual framework offers a fresh perspective on the classical divide between causal discovery and reasoning: sometimes, the main objective may be to foster scientific understanding by uncovering the qualitative causal structure underlying real-world systems; other times, causal discovery may only be a means to an end to support causal reasoning. Of particular interest in the context of actively selecting interventions is the setting in which we cannot directly intervene on variables whose causal effect on others we are interested in (see Fig. 4), which connects to concepts such as transportability and external validity [7, 57]. ABCI is also flexible in that it easily allows for incorporating available domain knowledge: if we know some aspects of the model a priori (as assumed in conventional causal reasoning) [53] or have access to a large observational sample (from which we can infer the MEC of DAGs) [1], we can encode this in our prior and only optimise over a smaller model class. The principled Bayesian nature of ABCI comes at a significant computational cost: most integrals are intractable and approximating them with Monte-Carlo sampling is computationally expensive and can introduce bias when resources are limited, though cf. [85] for recent efforts to address such intractability. We discuss the computational complexity of our implementation in more detail in Appx. E.3. On the other hand, in many real-world applications, such as in the context of biological networks, active interventions are possible but only at a significant cost [11, 53]. In such cases in particular, a careful and computationally-heavy experimental design approach as presented in the present work is warranted and could be easily amortised.

## Acknowledgments and Disclosure of Funding

We thank Paul K Rubenstein, Adrian Weller, and Bernhard Schölkopf for contributions to an early version of this work [80], and the anonymous reviewers for helpful feedback. This work was supported by: the German Federal Ministry of Education and Research (BMBF): Tübingen AI Center, FKZ: 01IS18039A, 01IS18039B; the Machine Learning Cluster of Excellence, EXC number 2064/1 – Project number 390727645; the European Research Council (ERC) under the European Union's Horizon 2020 research and innovation program grant agreement no. 815943; the Swiss National Science Foundation under NCCR Automation, grant agreement 51NF40 180545; and the Graz University of Technology LEAD project "Dependable Internet of Things in Adverse Environments".

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
