# Appendices

## Table of Contents

# A  Further Discussion of Related Work

In this section, we further discuss the most closely related prior works, which also consider a Bayesian active learning approach for causal discovery. These methods are summarised and contrasted with ABCI in Tab. 1. Similar to our approach, they also all assume acyclicity and causal sufficiency.

Table 1: Comparison of ABCI with closely related active Bayesian causal discovery methods in terms of the learning objective, that is, the causal target query, and the considered model class.

| Work | Target Query | Model Class |
| --- | --- | --- |
| Tong and Koller [79], Murphy [50] | causal graph $G$ | Conjugate Dirichlet-Multinomial |
| Cho et al. [11] | causal graph $G$ | Conjugate linear Gaussian-inverse-Gamma |
| Agrawal et al. [1] | some function $\phi(G)$ of the causal graph $G$ | Linear Gaussian |
| Tigas et al. [78] | causal graph $G$ and parameters of $f_i$ | Additive Gaussian noise with parametric neural network functions $f_i$ |
| GP-DiBS-ABCI (ours) | some function $q(\mathcal{M})$ of the full SCM $\mathcal{M}$ | Additive Gaussian noise with nonparametric functions $f_i$ modeled by GPs |

The early experimental design work by Tong and Koller [79] and Murphy [50] already investigated active causal discovery from a Bayesian perspective. They focused on the case in which all variables are multinomial to allow for tractable, closed-form posterior inference with a conjugate Dirichlet prior.

The setting with continuous variables was not explored from an active Bayesian causal discovery perspective until the work of Cho et al. [11], who consider the linear Gaussian case in the context of biological networks. Cho et al. [11] similarly use an inverse-Gamma prior to enable closed-form posterior inference. In these approaches, experiment selection targets the full causal graph. Agrawal et al. [1] extend the work of Cho et al. [11] by enabling the active learning of some function of the causal graph and handling interventional budget constraints.

Similarly to our approach, the concurrent work by Tigas et al. [78] models nonlinear causal relationships with additive Gaussian noise in the active learning setting. However, they are limited to targeting the full SCM for experiment design, which corresponds to our $q_{\mathrm{CML}}$ objective. In addition, their approach does not quantify the uncertainty in the functions conditional on a causal graph sampled from the graph posterior. In contrast, our nonparametric approach both directly models the epistemic uncertainty in the functions and mitigates the risk of model misspecification by jointly learning the kernel hyperparameters. Moreover, our method is Bayesian over the unknown noise variances, which are usually unknown in practice. It is unclear whether Tigas et al. [78] hand-specify a constant noise variance a priori, or whether they infer it jointly with the function parameters [78, cf. § 5.4.1].

Other related work by Shanmugam et al. [70] considers the problem of finding the minimal number of perfectly informative (w.r.t. conditional independences induced by the true underlying graph) multi-target interventions to fully identify the true causal graph. In contrast, we assume that only finitely many data points are available per experiment/intervention. Thus, we try to perform at each time step (possibly repeating) interventions to maximally reduce our uncertainty in the target causal query (which may be the causal graph). As another point of difference, we also optimise for the actual intervention value, whereas Shanmugam et al. [70] only optimise for the intervention targets.

# B Background on Gaussian processes

We use Gaussian Processes (GPs) to model mechanisms of non-root nodes $X_i$, i.e., we place a GP prior on $p(f_i \,|\, G)$. In the following, we give some background on GPs and how to compute probabilistic quantities thereof relevant to this work. For further information on GPs we refer the reader to Williams and Rasmussen [82].

A $\mathcal{GP}(m_i(\cdot), k_i^G(\cdot, \cdot))$ is a collection of random variables, any finite number of which have a joint Gaussian distribution, and is fully determined by its mean function $m_i(\cdot)$ and covariance function (or kernel) $k_i^G(\cdot, \cdot)$, where

$$m(\mathbf{x}) = \mathbb{E}[f(\mathbf{x})], \quad \text{and} \quad k(\mathbf{x}, \mathbf{x}') = \mathbb{E}[(f(\mathbf{x}) - m(\mathbf{x}))(f(\mathbf{x}') - m(\mathbf{x}'))]. \tag{B.1}$$

In our experiments, we choose the mean function $m_i(x) \equiv 0$ to be zero and a rational quadratic kernel

$$k_{RQ}(\mathbf{x}, \mathbf{x}') = \kappa_i^o \cdot \left(1 + \frac{1}{2\alpha}(\mathbf{x} - \mathbf{x}')^\top \kappa_i^l (\mathbf{x} - \mathbf{x}')\right)^{-\alpha} \tag{B.2}$$

as our covariance function. Here, $\alpha$ denotes a weighting parameter, $\kappa_i^o$ denotes an output scale parameter and $\kappa_i^l$ denotes a length scale parameter. For the weighting parameter, we use a default value of $\alpha = \log 2 \approx 0.693$. For $\kappa_i^l$ and $\kappa_i^o$ we choose priors according to Appx. D.4. In Section 4.1 we summarise both parameters as $\boldsymbol{\kappa}_i = (\kappa_i^o, \kappa_i^l)$.

In this work, we consider Gaussian additive noise models (see Eq. (4.1)). Hence, for a given non-root node $X_i$ in some graph $G$, we have

$$p(X_i \,|\, \mathbf{pa}_i^G, f_i, \sigma_i^2, G) = \mathcal{N}(X_i \,|\, f_i(\mathbf{pa}_i^G), \sigma_i^2) \tag{B.3}$$

where $\mathbf{pa}_i^G$ denotes the parents of $X_i$ in $G$. For some batch of collected data $\boldsymbol{x} = \{\boldsymbol{x}^n\}_{n=1}^N$, let $\boldsymbol{x}_i = (x_i^1, \dots x_i^N)^T$, $\mathbf{pa}_i^G = (\mathbf{pa}_i^{G,1}, \dots, \mathbf{pa}_i^{G,N})$, and $\boldsymbol{K}$ the Gram matrix with entries $K_{m,n} = k_{RQ}(\mathbf{pa}_i^{G,m}, \mathbf{pa}_i^{G,n})$. Then, we can compute the prior marginal log-likelihood, which is needed to compute $p(x^{1:t} \,|\, G)$, in closed form as

$$\log p(\boldsymbol{x}_i \,|\, \mathbf{pa}_i^G, \sigma_i^2, G) = \log \mathbb{E}_{f_i \,|\, G}\left[p(\boldsymbol{x}_i \,|\, \mathbf{pa}_i^G, f_i, \sigma_i^2, G)\right] \tag{B.4}$$

$$= -\frac{1}{2}\boldsymbol{x}_i^T (K + \sigma^2 I)^{-1} \boldsymbol{x}_i - \frac{1}{2}\log|K + \sigma^2 I| - \frac{N}{2}\log 2\pi. \tag{B.5}$$

To predict the function values $f_i(\widetilde{\mathbf{pa}}_i^G)$ at unseen test locations $\widetilde{\mathbf{pa}}_i^G = (\widetilde{\mathbf{pa}}_i^{G,1}, \dots, \widetilde{\mathbf{pa}}_i^{G,\tilde{N}})$ given previously observed data $\boldsymbol{x}$, let $\boldsymbol{K}^\dagger$ be the $(\tilde{N} \times N)$ covariance matrix with entries $K_{m,n}^\dagger = k_{RQ}(\widetilde{\mathbf{pa}}_i^{G,m}, \mathbf{pa}_i^{G,n})$ and $\tilde{\boldsymbol{K}}$ be the $(\tilde{N} \times \tilde{N})$ covariance matrix with entries $\tilde{K}_{m,n} = k_{RQ}(\widetilde{\mathbf{pa}}_i^{G,m}, \widetilde{\mathbf{pa}}_i^{G,n})$. Then, the predictive posterior is multivariate Gaussian

$$p(f_i(\widetilde{\mathbf{pa}}_i^G) \,|\, \widetilde{\mathbf{pa}}_i^G, \boldsymbol{x}, \sigma_i^2, G) = \mathcal{N}(\boldsymbol{\mu}_f, \boldsymbol{\Sigma}_f) \tag{B.6}$$

with mean

$$\boldsymbol{\mu}_f = \boldsymbol{K}^\dagger \left[\boldsymbol{K} + \sigma_i^2 I\right]^{-1} \boldsymbol{x}_i \tag{B.7}$$

and covariance

$$\boldsymbol{\Sigma}_f = \tilde{\boldsymbol{K}} - \boldsymbol{K}^\dagger \left[\boldsymbol{K} + \sigma_i^2 I\right]^{-1} \boldsymbol{K}^\dagger. \tag{B.8}$$

Finally, the marginal posterior over observations $\tilde{\boldsymbol{X}}_i$, which is needed to sample and evaluate candidate experiments in the experimental design process, is given by

$$p(\tilde{\boldsymbol{X}}_i \,|\, \widetilde{\mathbf{pa}}_i^G, \boldsymbol{x}, \sigma_i^2, G) = \mathcal{N}(\boldsymbol{\mu}_{X_i}, \boldsymbol{\Sigma}_{X_i}) \tag{B.9}$$

with mean

$$\boldsymbol{\mu}_{X_i} = \boldsymbol{\mu}_f \tag{B.10}$$

and covariance

$$\boldsymbol{\Sigma}_{X_i} = \boldsymbol{\Sigma}_f + \sigma_i^2 I. \tag{B.11}$$

# C  Derivation of the Information Gain Utility Functions

In the following, we provide the derivations for the expressions presented in Section 4.2.

## C.1  Information Gain for General Queries

We show that

$$\arg\max_{a_t} \; \mathrm{I}(Y; \boldsymbol{X}^t \,|\, \boldsymbol{x}^{1:t-1}) \;\; = \;\; \arg\max_{a_t} \; U(a_t) \tag{C.1}$$

for $U(a_t)$ given in Eq. (4.11).

**Proof.**  We write the mutual information in the following form

$$\mathrm{I}(Y; \boldsymbol{X}^t \,|\, \boldsymbol{x}^{1:t-1}) = H(Y \,|\, \boldsymbol{x}^{1:t-1}) + H(\boldsymbol{X}^t \,|\, \boldsymbol{x}^{1:t-1}) - H(Y, \boldsymbol{X}^t \,|\, \boldsymbol{x}^{1:t-1}). \tag{C.2}$$

In the above, we expand the joint entropy of experiment outcome and query as

$$H(Y, \boldsymbol{X}^t \,|\, \boldsymbol{x}^{1:t-1}) = -\mathbb{E}_{Y, \boldsymbol{X}^t \,|\, \boldsymbol{x}^{1:t-1}} \left[ \log p(Y, \boldsymbol{X}^t \,|\, \boldsymbol{x}^{1:t-1}) \right] \tag{C.3}$$

$$= -\mathbb{E}_{\mathcal{M} \,|\, \boldsymbol{x}^{1:t-1}} \left[ \mathbb{E}_{Y, \boldsymbol{X}^t \,|\, \mathcal{M}} \left[ \log p(Y, \boldsymbol{X}^t \,|\, \boldsymbol{x}^{1:t-1}) \right] \right] \tag{C.4}$$

$$= -\mathbb{E}_{\mathcal{M} \,|\, \boldsymbol{x}^{1:t-1}} \left[ \mathbb{E}_{Y, \boldsymbol{X}^t \,|\, \mathcal{M}} \left[ \log \mathbb{E}_{\mathcal{M}' \,|\, \boldsymbol{x}^{1:t-1}} \left[ p(Y \,|\, \mathcal{M}') \cdot p(\boldsymbol{X}^t \,|\, \mathcal{M}') \right] \right] \right] \tag{C.5}$$

for any query such that query and experiment outcome are conditionally independent given an SCM. This holds true, e.g., whenever $Y$ is a deterministic function of $\mathcal{M}$ such as $Y = q_{\mathrm{CD}}(\mathcal{M}) = G$.

The marginal entropy of the experiment outcome given previously observed data is

$$H(\boldsymbol{X}^t \,|\, \boldsymbol{x}^{1:t-1}) = -\mathbb{E}_{\boldsymbol{X}^t \,|\, \boldsymbol{x}^{1:t-1}} \left[ \log p(\boldsymbol{X}^t \,|\, \boldsymbol{x}^{1:t-1}) \right] \tag{C.6}$$

$$= -\mathbb{E}_{\mathcal{M} \,|\, \boldsymbol{x}^{1:t-1}} \left[ \mathbb{E}_{\boldsymbol{X}^t \,|\, \mathcal{M}} \left[ \log p(\boldsymbol{X}^t \,|\, \boldsymbol{x}^{1:t-1}) \right] \right] \tag{C.7}$$

$$= -\mathbb{E}_{\mathcal{M} \,|\, \boldsymbol{x}^{1:t-1}} \left[ \mathbb{E}_{\boldsymbol{X}^t \,|\, \mathcal{M}} \left[ \log \mathbb{E}_{\mathcal{M}' \,|\, \boldsymbol{x}^{1:t-1}} \left[ p(\boldsymbol{X}^t \,|\, \mathcal{M}') \right] \right] \right] \tag{C.8}$$

$$= -\mathbb{E}_{\mathcal{M} \,|\, \boldsymbol{x}^{1:t-1}} \left[ \mathbb{E}_{\boldsymbol{X}^t \,|\, \mathcal{M}} \left[ \log \mathbb{E}_{\boldsymbol{f}', \boldsymbol{\sigma}^{2'}, G' \,|\, \boldsymbol{x}^{1:t-1}} \left[ p(\boldsymbol{X}^t \,|\, \boldsymbol{f}', \boldsymbol{\sigma}^{2'}, G') \right] \right] \right] \tag{C.9}$$

$$= -\mathbb{E}_{\mathcal{M} \,|\, \boldsymbol{x}^{1:t-1}} \left[ \mathbb{E}_{\boldsymbol{X}^t \,|\, \mathcal{M}} \left[ \log \mathbb{E}_{G' \,|\, \boldsymbol{x}^{1:t-1}} \left[ p(\boldsymbol{X}^t \,|\, G', \boldsymbol{x}^{1:t-1}) \right] \right] \right] \tag{C.10}$$

$$= -\mathbb{E}_{\boldsymbol{f}, \boldsymbol{\sigma}^2, G \,|\, \boldsymbol{x}^{1:t-1}} \left[ \mathbb{E}_{\boldsymbol{X}^t \,|\, \boldsymbol{f}, \boldsymbol{\sigma}^2, G} \left[ \log \mathbb{E}_{G' \,|\, \boldsymbol{x}^{1:t-1}} \left[ p(\boldsymbol{X}^t \,|\, G', \boldsymbol{x}^{1:t-1}) \right] \right] \right] \tag{C.11}$$

$$= -\mathbb{E}_{G \,|\, \boldsymbol{x}^{1:t-1}} \left[ \mathbb{E}_{\boldsymbol{X}^t \,|\, G, \boldsymbol{x}^{1:t-1}} \left[ \log \mathbb{E}_{G' \,|\, \boldsymbol{x}^{1:t-1}} \left[ p(\boldsymbol{X}^t \,|\, G', \boldsymbol{x}^{1:t-1}) \right] \right] \right] \tag{C.12}$$

Finally, since the query posterior entropy $H(Y \,|\, \boldsymbol{x}^{1:t-1})$ does not depend on the candidate experiment $a_t$, we obtain

$$\arg\max_{a_t} \quad \mathrm{I}(Y; \boldsymbol{X}^t \,|\, \boldsymbol{x}^{1:t-1})$$

$$= \arg\max_{a_t} \quad H(Y \,|\, \boldsymbol{x}^{1:t-1}) + H(\boldsymbol{X}^t \,|\, \boldsymbol{x}^{1:t-1}) - H(Y, \boldsymbol{X}^t \,|\, \boldsymbol{x}^{1:t-1})$$

$$= \arg\max_{a_t} \quad H(\boldsymbol{X}^t \,|\, \boldsymbol{x}^{1:t-1}) - H(Y, \boldsymbol{X}^t \,|\, \boldsymbol{x}^{1:t-1}) \tag{C.13}$$

which, together with Eqs. (C.5) and (C.8), completes the proof. □

## C.2  Derivation of the Causal Discovery Utility Function

To derive $U_{\mathrm{CD}}(a)$, we note that $Y = q_{\mathrm{CD}}(\mathcal{M}) = G$, and hence the joint entropy of experiment outcome and query in Eq. (C.3) becomes

$$H(G, \boldsymbol{X}^t \,|\, \boldsymbol{x}^{1:t-1}) = -\mathbb{E}_{G, \boldsymbol{X}^t \,|\, \boldsymbol{x}^{1:t-1}} \left[ \log p(G, \boldsymbol{X}^t \,|\, \boldsymbol{x}^{1:t-1}) \right] \tag{C.14}$$

$$= -\mathbb{E}_{G, \boldsymbol{X}^t \,|\, \boldsymbol{x}^{1:t-1}} \left[ \log p(\boldsymbol{X}^t \,|\, G, \boldsymbol{x}^{1:t-1}) + \log p(G \,|\, \boldsymbol{x}^{1:t-1}) \right] \tag{C.15}$$

$$= -\mathbb{E}_{G, \boldsymbol{X}^t \,|\, \boldsymbol{x}^{1:t-1}} \left[ \log p(\boldsymbol{X}^t \,|\, G, \boldsymbol{x}^{1:t-1}) \right] + H(G \,|\, \boldsymbol{x}^{1:t-1}) \tag{C.16}$$

$$= -\mathbb{E}_{G \,|\, \boldsymbol{x}^{1:t-1}} \left[ \mathbb{E}_{\boldsymbol{X}^t \,|\, G, \boldsymbol{x}^{1:t-1}} \left[ \log p(\boldsymbol{X}^t \,|\, G, \boldsymbol{x}^{1:t-1}) \right] \right] + H(G \,|\, \boldsymbol{x}^{1:t-1}). \tag{C.17}$$

Substituting this into Eq. (C.2) yields

$$\mathrm{I}(G; \boldsymbol{X}^t \,|\, \boldsymbol{x}^{1:t-1}) \tag{C.18}$$
$$= H(\boldsymbol{X}^t \,|\, \boldsymbol{x}^{1:t-1}) + \mathbb{E}_{G \,|\, \boldsymbol{x}^{1:t-1}} \left[ \mathbb{E}_{\boldsymbol{X}^t \,|\, G, \boldsymbol{x}^{1:t-1}} \left[ \log p(\boldsymbol{X}^t \,|\, G, \boldsymbol{x}^{1:t-1}) \right] \right]. \tag{C.19}$$

By Eq. (C.12), we have

$$= \mathbb{E}_{G \,|\, \boldsymbol{x}^{1:t-1}} \left[ \mathbb{E}_{\boldsymbol{X}^t \,|\, G, \boldsymbol{x}^{1:t-1}} \left[ \log p(\boldsymbol{X}^t \,|\, G, \boldsymbol{x}^{1:t-1}) - \log \mathbb{E}_{G' \,|\, \boldsymbol{x}^{1:t-1}} \left[ p(\boldsymbol{X}^t \,|\, G', \boldsymbol{x}^{1:t-1}) \right] \right] \right] \tag{C.20}$$

which recovers the utility function $U_{\mathrm{CD}}(a)$ from Eq. (4.13).

## C.3 Derivation of the Causal Model Learning Utility Function

To derive $U_{\mathrm{CML}}(a)$ given $Y = q_{\mathrm{CML}}(\mathcal{M}) = \mathcal{M}$, the joint entropy of experiment outcome and query in Eq. (C.3) are given by

$$H(\mathcal{M}, \boldsymbol{X}^t \,|\, \boldsymbol{x}^{1:t-1}) = -\mathbb{E}_{\mathcal{M}, \boldsymbol{X}^t \,|\, \boldsymbol{x}^{1:t-1}} \left[ \log p(\mathcal{M}, \boldsymbol{X}^t \,|\, \boldsymbol{x}^{1:t-1}) \right] \tag{C.21}$$
$$= -\mathbb{E}_{\mathcal{M}, \boldsymbol{X}^t \,|\, \boldsymbol{x}^{1:t-1}} \left[ \log p(\boldsymbol{X}^t \,|\, \mathcal{M}, \boldsymbol{x}^{1:t-1}) + \log p(\mathcal{M} \,|\, \boldsymbol{x}^{1:t-1}) \right] \tag{C.22}$$
$$= -\mathbb{E}_{\mathcal{M}, \boldsymbol{X}^t \,|\, \boldsymbol{x}^{1:t-1}} \left[ \log p(\boldsymbol{X}^t \,|\, \mathcal{M}) \right] + H(\mathcal{M} \,|\, \boldsymbol{x}^{1:t-1}) \tag{C.23}$$
$$= -\mathbb{E}_{\mathcal{M} \,|\, \boldsymbol{x}^{1:t-1}} \left[ \mathbb{E}_{\boldsymbol{X}^t \,|\, \mathcal{M}} \left[ \log p(\boldsymbol{X}^t \,|\, \mathcal{M}) \right] \right] + H(\mathcal{M} \,|\, \boldsymbol{x}^{1:t-1}). \tag{C.24}$$

As previously, substituting this into Eq. (C.2) yields

$$\mathrm{I}(G; \boldsymbol{X}^t \,|\, \boldsymbol{x}^{1:t-1}) = H(\boldsymbol{X}^t \,|\, \boldsymbol{x}^{1:t-1}) + \mathbb{E}_{\mathcal{M} \,|\, \boldsymbol{x}^{1:t-1}} \left[ \mathbb{E}_{\boldsymbol{X}^t \,|\, \mathcal{M}} \left[ \log p(\boldsymbol{X}^t \,|\, \mathcal{M},) \right] \right] \tag{C.25}$$

and by Eq. (C.10), we have

$$= \mathbb{E}_{\mathcal{M} \,|\, \boldsymbol{x}^{1:t-1}} \left[ \mathbb{E}_{\boldsymbol{X}^t \,|\, \mathcal{M}} \left[ \log p(\boldsymbol{X}^t \,|\, \mathcal{M}) - \log \mathbb{E}_{G' \,|\, \boldsymbol{x}^{1:t-1}} \left[ p(\boldsymbol{X}^t \,|\, G', \boldsymbol{x}^{1:t-1}) \right] \right] \right] \tag{C.26}$$

which recovers the utility $U_{\mathrm{CML}}(a)$ from Eq. (4.14).

For our concrete modeling choices we can further simplify this utility. Let $\mathbf{Anc}_i^{\mathcal{M}}$ and $\mathbf{Pa}_i^{\mathcal{M}}$ denote the ancestor and parent sets of node $X_i$ in $\mathcal{M}$. Then,

$$\mathbb{E}_{\mathcal{M} \,|\, \boldsymbol{x}^{1:t-1}} \left[ \mathbb{E}_{\boldsymbol{X}^t \,|\, \mathcal{M}} \left[ \log p(\boldsymbol{X}^t \,|\, \mathcal{M}) \right] \right] \tag{C.27}$$

$$= \mathbb{E}_{\mathcal{M} \,|\, \boldsymbol{x}^{1:t-1}} \left[ \mathbb{E}_{\boldsymbol{X}^t \,|\, \mathcal{M}} \left[ \log \prod_{i \notin \mathcal{I}^t} p^{\mathrm{do}(a_t)}(\boldsymbol{X}_i^t \,|\, \mathbf{pa}_i^{\mathcal{M}}, \mathcal{M}) \right] \right] \tag{C.28}$$

$$= \mathbb{E}_{\mathcal{M} \,|\, \boldsymbol{x}^{1:t-1}} \left[ \mathbb{E}_{\boldsymbol{X}^t \,|\, \mathcal{M}} \left[ \sum_{i \notin \mathcal{I}^t} \log p^{\mathrm{do}(a_t)}(\boldsymbol{X}_i^t \,|\, \mathbf{pa}_i^{\mathcal{M}}, \mathcal{M}) \right] \right] \tag{C.29}$$

$$= \mathbb{E}_{\mathcal{M} \,|\, \boldsymbol{x}^{1:t-1}} \left[ \sum_{i \notin \mathcal{I}^t} \mathbb{E}_{\boldsymbol{X}^t \,|\, \mathcal{M}} \left[ \log p^{\mathrm{do}(a_t)}(\boldsymbol{X}_i^t \,|\, \mathbf{pa}_i^{\mathcal{M}}, \mathcal{M}) \right] \right] \tag{C.30}$$

$$= \mathbb{E}_{\mathcal{M} \,|\, \boldsymbol{x}^{1:t-1}} \left[ \sum_{i \notin \mathcal{I}^t} \mathbb{E}_{\mathbf{Anc}_i^{\mathcal{M}} \,|\, \mathrm{do}(a_t), \mathcal{M}} \left[ \mathbb{E}_{\boldsymbol{X}_i^t \,|\, \mathbf{pa}_i^{\mathcal{M}}, \mathrm{do}(a_t), \mathcal{M}} \left[ \log p^{\mathrm{do}(a_t)}(\boldsymbol{X}_i^t \,|\, \mathbf{pa}_i^{\mathcal{M}}, \mathcal{M}) \right] \right] \right]. \tag{C.31}$$

Since our root nodes and GPs assume an additive Gaussian noise model, the innermost expectation amounts to the negative entropy the Gaussian noise variable, i.e.,

$$\mathbb{E}_{\boldsymbol{X}_i^t \,|\, \mathbf{pa}_i^{\mathcal{M}}, \mathrm{do}(a_t), \mathcal{M}} \left[ \log p^{\mathrm{do}(a_t)}(\boldsymbol{X}_i^t \,|\, \mathbf{pa}_i^{\mathcal{M}}, \mathcal{M}) \right] = -\frac{N_t}{2} \log(2\pi\sigma_i^2 e). \tag{C.32}$$

As we further assume a homoscedastic noise model for our GPs, Eq. (C.31) reduces to

$$\mathbb{E}_{\mathcal{M} \mid \boldsymbol{x}^{1:t-1}} \left[ \sum_{i \notin \mathcal{I}^t} -\frac{N_t}{2} \log(2\pi\sigma_i^2 e) \right] \tag{C.33}$$

$$= -\mathbb{E}_{\boldsymbol{f}, \boldsymbol{\sigma}^2, G \mid \boldsymbol{x}^{1:t-1}} \left[ \sum_{i \notin \mathcal{I}^t} \frac{N_t}{2} \log(2\pi\sigma_i^2 e) \right] \tag{C.34}$$

$$= -\mathbb{E}_{G \mid \boldsymbol{x}^{1:t-1}} \left[ \mathbb{E}_{\boldsymbol{\sigma}^2 \mid G, \boldsymbol{x}^{1:t-1}} \left[ \sum_{i \notin \mathcal{I}^t} \frac{N_t}{2} \log(2\pi\sigma_i^2 e) \right] \right] \tag{C.35}$$

$$= -\mathbb{E}_{G \mid \boldsymbol{x}^{1:t-1}} \left[ \sum_{i \notin \mathcal{I}^t} \mathbb{E}_{\sigma_i^2 \mid G, \boldsymbol{x}^{1:t-1}} \left[ \frac{N_t}{2} \log(2\pi\sigma_i^2 e) \right] \right], \tag{C.36}$$

which can be approximated by nested Monte Carlo estimation. For non-root nodes we approximate the inner expectation with a single point estimate (cf. Section 4.1). For root nodes we can compute the inner expectation in closed form as

$$\mathbb{E}_{\sigma_i^2 \mid G, \boldsymbol{x}^{1:t-1}} \left[ \frac{N_t}{2} \log(2\pi\sigma_i^2 e) \right] = \frac{N_t}{2} \left( \log(2\pi e) - \psi(\alpha_i^t) + \log \beta_i^t \right) \tag{C.37}$$

where $\alpha_i^t, \beta_i^t$ are the parameters of the inverse-gamma noise posterior $\sigma_i^2 \sim \Gamma^{-1}(\sigma_i^2 \mid \alpha_i^t, \beta_i^t)$ (see Appx. D.3) and $\psi(\cdot)$ is the digamma function.

**Proof** (adapted from [74]). We need to show that

$$\mathbb{E}_{\sigma^2} \left[ \log(\sigma^2) \right] = -\psi(\alpha) + \log \beta \tag{C.38}$$

where the noise variance $\sigma^2$ follows an inverse-gamma density

$$\sigma^2 \sim \Gamma^{-1}(\sigma^2 \mid \alpha, \beta) = \frac{\beta^\alpha}{\Gamma(\alpha)} \cdot (\sigma^2)^{-\alpha-1} \cdot e^{-\frac{\beta}{\sigma^2}}. \tag{C.39}$$

By substituting $y = \log \sigma^2$ we get

$$y \sim p(y \mid \alpha, \beta) = \frac{\beta^\alpha}{\Gamma(\alpha)} \cdot e^{-\alpha y} \cdot e^{-\beta e^{-y}}. \tag{C.40}$$

Now note that

$$\int_{-\infty}^{\infty} p(y \mid \alpha, \beta) dy = 1 \tag{C.41}$$

and hence

$$\frac{\Gamma(\alpha)}{\beta^\alpha} = \int_{-\infty}^{\infty} e^{-\alpha y} \cdot e^{-\beta e^{-y}} dy. \tag{C.42}$$

By differentiating the latter integrand w.r.t. $\alpha$ we get

$$\frac{d}{d\alpha} \left( e^{-\alpha y} \cdot e^{-\beta e^{-y}} \right) = (-y) e^{-\alpha y} \cdot e^{-\beta e^{-y}} = (-y) \cdot p(y \mid \alpha, \beta) \cdot \frac{\Gamma(\alpha)}{\beta^\alpha}. \tag{C.43}$$

Bringing the parts together we obtain

$$\mathbb{E}_{\sigma^2}\left[\log(\sigma^2)\right] = \qquad \mathbb{E}_y\left[y\right] \tag{C.44}$$

$$= \qquad \int_{-\infty}^{\infty} y \cdot p(y \mid \alpha, \beta) dy \tag{C.45}$$

$$\overset{Eq. \ (C.43)}{=} \quad -\frac{\beta^\alpha}{\Gamma(\alpha)} \int_{-\infty}^{\infty} \frac{d}{d\alpha}\left(e^{-\alpha y} \cdot e^{-\beta e^{-y}}\right) dy \tag{C.46}$$

$$= \qquad -\frac{\beta^\alpha}{\Gamma(\alpha)} \frac{d}{d\alpha}\left(\int_{-\infty}^{\infty} e^{-\alpha y} \cdot e^{-\beta e^{-y}} dy\right) \tag{C.47}$$

$$\overset{Eq. \ (C.42)}{=} \quad -\frac{\beta^\alpha}{\Gamma(\alpha)} \frac{d}{d\alpha}\left(\frac{\Gamma(\alpha)}{\beta^\alpha}\right) \tag{C.48}$$

$$= \qquad -\frac{\beta^\alpha}{\Gamma(\alpha)}\left(\beta^{-\alpha} \cdot \frac{d}{d\alpha}\Gamma(\alpha) - \Gamma(\alpha) \cdot \beta^{-\alpha} \cdot \log\beta\right) \tag{C.49}$$

$$= \qquad -\psi(\alpha) + \log\beta, \tag{C.50}$$

which completes the proof. $\qquad\square$

In summary, in our instance of GP-DIBS-ABCI we estimate the causal model learning utility as

$$U_{\text{CML}}(a_t) = -\mathbb{E}_{G \mid \boldsymbol{x}^{1:t-1}}\Bigg[ \sum_{i \in \mathbf{R}(G) \setminus \mathcal{I}^t} \frac{N_t}{2}\left(\log(2\pi e) - \psi(\alpha_i^t) + \log\beta_i^t\right) +$$

$$\sum_{i \in \mathbf{NR}(G) \setminus \mathcal{I}^t} \mathbb{E}_{\sigma_i^2 \mid G, \boldsymbol{x}^{1:t-1}}\left[\frac{N_t}{2}\log(2\pi\sigma_i^2 e)\right] +$$

$$\mathbb{E}_{\boldsymbol{X}^t \mid G, \boldsymbol{x}^{1:t-1}}\left[\log \mathbb{E}_{G' \mid \boldsymbol{x}^{1:t-1}}\left[p(\boldsymbol{X}^t \mid G', \boldsymbol{x}^{1:t-1})\right]\right]\Bigg] \tag{C.51}$$

## C.4   Derivation of the Causal Reasoning Utility Function

We derive the utility function $U_{\text{CR}}(a)$ in Eq. (4.12) for the query $Y = X_j^{\text{do}(X_i=\psi)}$ with $\psi \sim p(\psi)$ a distribution over intervention values. Starting with the joint entropy in Eq. (C.3) we marginalise over graphs (instead of SCMs) to exploit that we can sample from and evaluate $p(\boldsymbol{X} \mid G, \boldsymbol{x}^{1:t-1})$ in closed form by using GPs:

$$-H(Y, \boldsymbol{X}^t \mid \boldsymbol{x}^{1:t-1})$$

$$= \mathbb{E}_{Y, \boldsymbol{X}^t \mid \boldsymbol{x}^{1:t-1}}\left[\log p(Y, \boldsymbol{X}^t \mid \boldsymbol{x}^{1:t-1})\right] \tag{C.52}$$

$$= \mathbb{E}_{G \mid \boldsymbol{x}^{1:t-1}}\left[\mathbb{E}_{Y, \boldsymbol{X}^t \mid G, \boldsymbol{x}^{1:t-1}}\left[\log \mathbb{E}_{G' \mid \boldsymbol{x}^{1:t-1}}\left[p(Y, \boldsymbol{X}^t \mid G', \boldsymbol{x}^{1:t-1})\right]\right]\right] \tag{C.53}$$

$$= \mathbb{E}_{G \mid \boldsymbol{x}^{1:t-1}}\Big[\mathbb{E}_{\boldsymbol{X}^t \mid G, \boldsymbol{x}^{1:t-1}}\Big[\mathbb{E}_{Y \mid \boldsymbol{X}^t, G, \boldsymbol{x}^{1:t-1}}\Big[$$
$$\log \mathbb{E}_{G' \mid \boldsymbol{x}^{1:t-1}}\left[p(Y \mid \boldsymbol{X}^t, G', \boldsymbol{x}^{1:t-1}) \cdot p(\boldsymbol{X}^t \mid G', \boldsymbol{x}^{1:t-1})\right]\Big]\Big]\Big] \tag{C.54}$$

To estimate $\mathbb{E}_{Y \mid \boldsymbol{X}^t, G, \boldsymbol{x}^{1:t-1}}[\cdot]$ we first sample intervention values $\psi \sim p(\psi)$ and then sample from the respective interventional densities $p^{\text{do}(X_i=\psi)}(X_j \mid \boldsymbol{X}^t, G, \boldsymbol{x}^{1:t-1})$ induced by candidate SCMs with graph $G$. Thus, the expectation becomes $\mathbb{E}_\psi\left[\mathbb{E}_{X_j \mid \boldsymbol{X}^t, G, \boldsymbol{x}^{1:t-1}}^{\text{do}(X_i=\psi)}[\cdot]\right]$. To evaluate $p(Y \mid \boldsymbol{X}^t, G', \boldsymbol{x}^{1:t-1})$ we estimate $p^{\text{do}(X_i=\psi)}(X_j \mid \boldsymbol{X}^t, G', \boldsymbol{x}^{1:t-1})$ as described in Appx. D.1. The joint entropy therefore becomes

$$-H(Y, \boldsymbol{X}^t \mid \boldsymbol{x}^{1:t-1}) = \mathbb{E}_{G \mid \boldsymbol{x}^{1:t-1}}\Big[\mathbb{E}_{\boldsymbol{X}^t \mid G, \boldsymbol{x}^{1:t-1}}\Big[\mathbb{E}_\psi\Big[\mathbb{E}_{X_j \mid \boldsymbol{X}^t, G, \boldsymbol{x}^{1:t-1}}^{\text{do}(X_i=\psi)}\Big[ \tag{C.55}$$
$$\log \mathbb{E}_{G' \mid \boldsymbol{x}^{1:t-1}}\left[p^{\text{do}(X_i=\psi)}(X_j \mid \boldsymbol{X}^t, G', \boldsymbol{x}^{1:t-1}) \cdot p(\boldsymbol{X}^t \mid G', \boldsymbol{x}^{1:t-1})\right]\Big]\Big]\Big]\Big]$$

By substituting Eqs. (C.13) and (D.1) into Eq. (C.12) we obtain the causal reasoning utility in Eq. (4.12).

# D  Approximate Inference and Experimental Details

In this section, we provide details about our approximate inference and estimation procedures, including the estimation of the marginal interventional likelihoods in Section D.1 and prior choices in Sections D.2 — D.4. We also provide details on DiBS for approximate graph posterior inference in Section D.5, the estimation of the information gain utilities in Section D.6, and our use of Bayesian Optimisation for experimental design in Section D.7.

## D.1  Estimating Posterior Marginal Interventional Likelihoods

In the following, we show how we estimate (posterior) marginal interventional likelihoods $p^{\mathrm{do}(x_j)}(\boldsymbol{x}_i \mid \boldsymbol{x}^{1:t})$. Let $\mathbf{Anc}_i^G$ and $\mathbf{Pa}_i^G$ denote the ancestor and parent sets of node $X_i$ in $G$. Then, the marginal interventional likelihood is given by

$$
p^{\mathrm{do}(x_j)}(\boldsymbol{x}_i \mid \boldsymbol{x}^{1:t})
$$

$$
= \mathbb{E}_{\mathcal{M} \mid \boldsymbol{x}^{1:t}} \left[ p^{\mathrm{do}(x_j)}(\boldsymbol{x}_i \mid \mathcal{M}) \right] \tag{D.1}
$$

$$
= \mathbb{E}_{\boldsymbol{f}, \boldsymbol{\sigma}^2, G \mid \boldsymbol{x}^{1:t}} \left[ p^{\mathrm{do}(x_j)}(\boldsymbol{x}_i \mid \boldsymbol{f}, \boldsymbol{\sigma}^2, G) \right] \tag{D.2}
$$

$$
= \mathbb{E}_{\boldsymbol{f}, \boldsymbol{\sigma}^2, G \mid \boldsymbol{x}^{1:t}} \left[ \mathbb{E}_{\mathbf{Anc}_i^G \mid \mathrm{do}(x_j), \boldsymbol{f}, \boldsymbol{\sigma}^2, G} \left[ p^{\mathrm{do}(x_j)}(\boldsymbol{x}_i \mid \mathbf{anc}_i^G, \boldsymbol{f}, \boldsymbol{\sigma}^2, G) \right] \right] . \tag{D.3}
$$

Given that $X_i$ is independent of it's non-descendants given its parents, we obtain

$$
= \mathbb{E}_{\boldsymbol{f}, \boldsymbol{\sigma}^2, G \mid \boldsymbol{x}^{1:t}} \left[ \mathbb{E}_{\mathbf{Anc}_i^G \mid \mathrm{do}(x_j), \boldsymbol{f}, \boldsymbol{\sigma}^2, G} \left[ p^{\mathrm{do}(x_j)}(\boldsymbol{x}_i \mid \mathbf{pa}_i^G, f_i, \sigma_i^2, G) \right] \right] \tag{D.4}
$$

$$
= \mathbb{E}_{G \mid \boldsymbol{x}^{1:t}} \left[ \mathbb{E}_{\boldsymbol{f}, \boldsymbol{\sigma}^2 \mid G, \boldsymbol{x}^{1:t}} \left[ \mathbb{E}_{\mathbf{Anc}_i^G \mid \mathrm{do}(x_j), \boldsymbol{f}, \boldsymbol{\sigma}^2, G} \left[ p^{\mathrm{do}(x_j)}(\boldsymbol{x}_i \mid \mathbf{pa}_i^G, f_i, \sigma_i^2, G) \right] \right] \right] . \tag{D.5}
$$

Given that $p(\boldsymbol{f}, \boldsymbol{\sigma}^2 \mid G, \boldsymbol{x}^{1:t})$ factorises and $\mathbf{Anc}_i^G$ are independent of mechanisms and noise variances $\boldsymbol{f}, \boldsymbol{\sigma}^2$ of the non-ancestors of $X_i$, we have

$$
= \mathbb{E}_{G \mid \boldsymbol{x}^{1:t}} \left[ \mathbb{E}_{\boldsymbol{f}_{\mathbf{Anc}_i^G}, \boldsymbol{\sigma}^2_{\mathbf{Anc}_i^G} \mid G, \boldsymbol{x}^{1:t}} \left[ \mathbb{E}_{\mathbf{Anc}_i^G \mid \mathrm{do}(x_j), \boldsymbol{f}_{\mathbf{Anc}_i^G}, \boldsymbol{\sigma}^2_{\mathbf{Anc}_i^G}, G} \left[ \phantom{xxx} \right. \right. \right.
$$
$$
\left. \left. \left. \mathbb{E}_{f_i, \sigma_i^2 \mid G, \boldsymbol{x}^{1:t}} \left[ p^{\mathrm{do}(x_j)}(\boldsymbol{x}_i \mid \mathbf{pa}_i^G, f_i, \sigma_i^2, G) \right] \right] \right] \right] . \tag{D.6}
$$

Finally, marginalising out the functions and noise variances, we obtain

$$
= \mathbb{E}_{G \mid \boldsymbol{x}^{1:t}} \left[ \mathbb{E}_{\boldsymbol{f}_{\mathbf{Anc}_i^G}, \boldsymbol{\sigma}^2_{\mathbf{Anc}_i^G} \mid G, \boldsymbol{x}^{1:t}} \left[ \mathbb{E}_{\mathbf{Anc}_i^G \mid \mathrm{do}(x_j), \boldsymbol{f}_{\mathbf{Anc}_i^G}, \boldsymbol{\sigma}^2_{\mathbf{Anc}_i^G}, G} \left[ p^{\mathrm{do}(x_j)}(\boldsymbol{x}_i \mid \mathbf{pa}_i^G, G) \right] \right] \right] \tag{D.7}
$$

$$
= \mathbb{E}_{G \mid \boldsymbol{x}^{1:t}} \left[ \mathbb{E}_{\mathbf{Anc}_i^G \mid \mathrm{do}(x_j), G} \left[ p^{\mathrm{do}(x_j)}(\boldsymbol{x}_i \mid \mathbf{pa}_i^G, G) \right] \right] \tag{D.8}
$$

$$
= \mathbb{E}_{G \mid \boldsymbol{x}^{1:t}} \left[ \mathbb{E}_{\mathbf{Anc}_i^G \mid \mathrm{do}(x_j), G} \left[ p(\boldsymbol{x}_i \mid \mathbf{pa}_i^G, G) \big|_{X_j = x_j} \right] \right] . \tag{D.9}
$$

We use Monte Carlo estimation to approximate the outer expectation of this quantity according to Eq. (4.10). To approximate the inner expectation by performing ancestral sampling from the interventional density $p^{\mathrm{do}(x_j)}(\boldsymbol{X} \mid G)$, where we use 50 samples when estimating the $U_{\mathrm{CR}}$ utility in Equation Eq. (4.12) and 200 samples when estimating the metrics described in Appx. F.

## D.2  Sampling Ground Truth Graphs

When generating ground truth SCMs for evaluation, we sample causal graphs according to two random graph models. First, we sample scale-free graphs using the preferential attachment process presented by Barabási and Albert [6]. We use the `networkx.generators.barabasi_albert_graph` implementation provided in the NetworkX [29] Python package and interpret the returned, undirected graph as a DAG by only considering the upper-triangular part of its adjacency matrix. Before permuting the node labels, we generate graphs with in-degree 2 for nodes $\{X_i\}_{i=3}^d$ whereas $X_1$ and $X_2$ are always

root nodes. In addition, we consider Erdös-Renyi random graphs [20], where edges are sampled independently with probability $p = \frac{4}{d-1}$. After sampling edges, we choose a random ordering and discard any edges that disobey this ordering to obtain a DAG. Our choice of $p$ yields an expected degree of 2. Unlike Lorch et al. [45], we do not provide our model with any kind of prior information on the graph structure.

### D.3 Normal-Inverse-Gamma Prior for Root Nodes

We use a conjugate normal-inverse-gamma (N-$\Gamma^{-1}$) prior

$$p(f_i, \sigma_i^2 \,|\, G) = \text{N-}\Gamma^{-1}(\mu_i, \lambda_i, \alpha_i^R, \beta_i^R) \tag{D.10}$$

as the joint prior over functions and noise parameters for root nodes in $G$ (see Section 4 and Fig. 2). In our experiments, we use $\mu_i = 0$, $\lambda_i = 0.1$, $\alpha_i^R = 50$ and $\beta_i^R = 25$. When generating ground truth SCMs, we draw one sample for $(f_i^\star, \sigma_i^{2,\star})$ from this prior for all $i$ and leave it fixed thereafter. Closed-form expressions for the (posterior) marginal likelihood can be found, e.g., in [51].

### D.4 Gamma Priors for GP Hyperparameters of Non-Root Nodes

We model non-root node mechanisms with GPs (see Section 4.1), where each GP has a set of hyperparameters $(\boldsymbol{\kappa}_i, \sigma_i^2)$ where $\boldsymbol{\kappa}_i = (\kappa_i^l, \kappa_i^o)$ includes a length scale and output scale parameter, respectively, and where $\sigma_i^2$ denotes the variance of the Gaussian noise variable $U_i$. In our experiments, we use $p(\sigma_i^2 \,|\, G) = \text{Gamma}(\alpha = 50, \beta = 500)$, $p(\kappa_i^o \,|\, G) = \text{Gamma}(\alpha = 100, \beta = 10)$ and $p(\kappa_i^l \,|\, G) = \text{Gamma}(\alpha = 30 \cdot |\mathbf{Pa}_i^G|, \beta = 30)$, where $|\mathbf{Pa}_i^G|$ denotes the size of the parent set of $X_i$ in $G$.

### D.5 DiBS for Approximate Posterior Graph Inference

DiBS [45] introduces a probabilistic latent space representation for DAGs to allow for efficient posterior inference in continuous space. Specifically, given some latent particle $\boldsymbol{z} \in \mathbb{R}^{d \times d \times 2}$ we can define an edge-wise generative model

$$p(G \,|\, \boldsymbol{z}) = \prod_{i=1}^{d} \prod_{\substack{j=1 \\ j \neq i}}^{d} p(G_{i,j} \,|\, \boldsymbol{z}) \tag{D.11}$$

where $G_{i,j} \in \{0, 1\}$ indicates the absence/presence of an edge from $X_i$ to $X_j$ in $G$, and a prior distribution

$$p(\boldsymbol{Z}) \propto \exp(-\beta \, \mathbb{E}_{G \,|\, \boldsymbol{Z}}[h(G)]) \prod_{i,j,k} \mathcal{N}(z_{i,j,k} \,|\, 0, 1) \tag{D.12}$$

where $h(G)$ is a scoring function quantifying the "degree of cyclicity" of $G$. $\beta$ is a temperature parameter weighting the influence of the expected cyclicity in the prior. Lorch et al. [45] propose to use Stein Variational Gradient Descent [44] for approximate inference of $p(\boldsymbol{Z} \,|\, \boldsymbol{x}^{1:t})$. SVGD maintains a fixed set of particles $\boldsymbol{z} = \{\boldsymbol{z}_m\}_{m=1}^M$ and updates them using the posterior score $\nabla \log p(\boldsymbol{z} \,|\, \boldsymbol{x}^{1:t}) = \nabla \log p(\boldsymbol{z}) + \nabla \log p(\boldsymbol{x}^{1:t} \,|\, \boldsymbol{z})$. In our experiments, we use $K = 5$ latent particles. For the estimation of expectations as in Eq. (4.10), we use $K = 40$ MC graph samples unless otherwise stated, hence, a total of $M \cdot K = 200$ graphs, and we use the DiBS+ particle weighting. In contrast to the original DiBS version, we do not use the annealing parameter $\alpha$ to force the mass of $p(G|\boldsymbol{z})$ onto a single graph during training. For further details on the method and its implementation, we refer to the original publication [45] and the provided code.

### D.6 Estimation of the Information Gain Utility Functions

When estimating the information gain utilities (see § 4.2 and Appx. C), we keep the set of Monte Carlo samples from the SCM posterior $p(\mathcal{M} \,|\, \boldsymbol{x}^{1:t})$ fixed for all evaluations of the chosen utility during a given experiment design phase at time $t$, i.e., during the optimisation for all candidate intervention sets and intervention targets. In our experiments, for the $U_{\text{CD}}$ and $U_{\text{CML}}$ utilities we sample 5 and 30 graphs to approximate the outer and inner expectations w.r.t. the posterior graphs, respectively.

We sample 100 hypothetical experiment outcomes with given batch size from $p(\boldsymbol{X}^t \,|\, G, \boldsymbol{x}^{1:t})$ to approximate the expectation $\mathbb{E}_{\boldsymbol{X}^t \,|\, G, \boldsymbol{x}^{1:t}}[\cdot]$.

For the $U_{\mathrm{CR}}$ utility we sample 3 and 9 graphs to approximate the outer and inner expectations w.r.t. the posterior graphs, respectively. We sample 50 hypothetical experiment outcomes with given batch size from $p(\boldsymbol{X}^t \,|\, G, \boldsymbol{x}^{1:t})$ to approximate expectations of the form $\mathbb{E}_{\boldsymbol{X}^t \,|\, G, \boldsymbol{x}^{1:t}}[\cdot]$. To approximate the expectations $\mathbb{E}_\psi \big[\mathbb{E}^{\mathrm{do}(X_i = \psi)}_{X_j \,|\, \boldsymbol{X}^t, G, \mathcal{D}}[\cdot]\big]$ we sample 5 intervention values from $p(\psi)$ and draw 3 samples from $p^{\mathrm{do}(X_i = \psi)}(X_j \,|\, \boldsymbol{X}^t, G, \mathcal{D})$ for each intervention value.

### D.7  Bayesian Optimisation for Experimental Design

In order to find the optimal experiment $a_t^\star = (\mathcal{I}^\star, \boldsymbol{x}_{\mathcal{I}}^\star)$ at time $t$, we compute the optimal intervention value $\boldsymbol{x}_{\mathcal{I}}^\star \in \arg\max_{\boldsymbol{x}} U(\mathcal{I}, \boldsymbol{x})$ for each candidate intervention target set $\mathcal{I}$ (see Eq. (4.15)). As the evaluation of our proposed utility functions $U(a)$ is expensive, we require an efficient approach for finding optimal intervention values using as few function evaluations as possible. Following von Kügelgen et al. [80], we employ *Bayesian optimisation* (BO) [46, 47] for this task and model our uncertainty in $U(\mathcal{I}, \boldsymbol{x})$ given previous evaluations $\mathcal{D}_{BO} = \{(\boldsymbol{x}_l, U(\mathcal{I}, \boldsymbol{x}_l))\}_{l=1}^k$ with a GP. We select a new candidate solution according to the GP-UCB acquisition function [76],

$$\mathbf{x}_{k+1} = \arg\max_{\mathbf{x}} \mu_k(\mathbf{x}) + \gamma \sigma_k(\mathbf{x}), \tag{D.13}$$

where $\mu_k(\mathbf{x})$ and $\sigma_k(\mathbf{x})$ correspond to the mean and standard deviation of the GP predictive distribution $p(U(\mathcal{I}, \boldsymbol{x}) \,|\, \mathcal{D}_{BO})$ (see Appx. B). We then evaluate $U(\mathcal{I}, \boldsymbol{x}_{k+1})$ at the selected $\boldsymbol{x}_{k+1}$ and repeat. The scalar factor $\gamma$ trades off exploitation with exploration. In our experiments, we set $\gamma = 1$ and run the GP-UCB algorithm 8 times for each candidate set of intervention targets.

**Algorithm 2:** Particle Resampling

---

**Input:** set of latent particles $\boldsymbol{z} = \{\boldsymbol{z}_k\}_{k=1}^K$
**Output:** set of resampled latent particles $\tilde{\boldsymbol{z}} = \{\tilde{\boldsymbol{z}}_k\}_{k=1}^K$

$\tilde{z} \leftarrow \varnothing$          ▷initialise set of resampled particles
$N_{max} \leftarrow \left\lceil \frac{K}{4} \right\rceil$          ▷max. number of particles to keep
$\{w_k\}_{k=1}^K \leftarrow \left\{ \frac{p(\boldsymbol{z}_k \mid \boldsymbol{x}^{1:t})\, \tilde{p}(\boldsymbol{z}_k)}{\sum_k p(\boldsymbol{z}_k \mid \boldsymbol{x}^{1:t})\, \tilde{p}(\boldsymbol{z}_k)} \right\}$          ▷compute particle weights
$n_{kept} \leftarrow 0$
**for** $w_k$ **in** sort_descending($\{w_k\}_{k=1}^K$) **do**
     **if** $n_{kept} < N_{max}$ **and** $w_k > 0.01$ **then**
         $\tilde{\boldsymbol{z}} \leftarrow \tilde{\boldsymbol{z}} \cup \{\boldsymbol{z}_k\}\ n_{kept} \leftarrow n_{kept} + 1$
     **end**
     **else**
         $\boldsymbol{z}_{new} \sim p(\boldsymbol{Z})$
         $\tilde{\boldsymbol{z}} \leftarrow \tilde{\boldsymbol{z}} \cup \{\boldsymbol{z}_{new}\}$
     **end**
**end**

---

# E   Implementation Details

In this section, we give details about our implementation, including our particle resampling procedure in Section E.1, the sharing and caching of priors in Section E.2, a discussion of the computational complexity of our implementation in Section E.3, and finally some information on our code framework and computing resources in Section E.4. Our implementation is available at https://www.github.com/chritoth/active-bayesian-causal-inference.

## E.1   Particle Resampling

As described in Alg. 1, we resample latent particles $\boldsymbol{z} = \{\boldsymbol{z}_k\}_{k=1}^K$ according to a predefined schedule instead of sampling new particles from the particle prior $p(\boldsymbol{Z})$ after each epoch. Although sampling new particles would allow for higher diversity in the graph Monte Carlo samples and their respective mechanisms, it also entails a higher computational burden as the caching of mechanism marginal log-likelihoods is not as effective anymore. On the other hand, keeping a subset of the inferred particles is efficient, because once we have inferred a "good" particle $\boldsymbol{z}_k$ that supposedly has a high posterior density $p(\boldsymbol{z}_k \mid \boldsymbol{x}^{1:t})$ it would be wasteful to discard the particle only to infer a similar particle again. Empirically, we found that keeping particles depending on their unnormalized posterior densities according to Alg. 2 does not diminish inference quality while increasing computational efficiency. In our experiments, we chose the following resampling schedule:

$$ r_t = \begin{cases} 1 & \text{if} \quad t \in \{1, 2, 3, 4, 5, 6, 9\} \\ 1 & \text{if} \quad t \bmod 5 = 0 \\ 0 & \text{otherwise.} \end{cases} $$

## E.2   Shared Priors and Caching of Marginal Likelihoods

We share priors for mechanisms and noise $p(f_i, \sigma_i \mid G)$, as well as for GP hyperparameters $p(\boldsymbol{\kappa}_i \mid G)$, across all graphs $G$ that induce the same parent set $\mathbf{Pa}_i^G$. Consequently, not only the posteriors $p(f_i, \sigma_i \mid G, \boldsymbol{x}^{1:t})$ and $p(\boldsymbol{\kappa}_i \mid G, \boldsymbol{x}^{1:t})$, but also the GP marginal likelihoods $p(\boldsymbol{x}_i^{1:t} \mid G)$ and GP predictive marginal likelihoods $p(\boldsymbol{x}_i^{t+1} \mid G, \boldsymbol{x}^{1:t})$ can be shared across graphs with identical parent sets for node $X_i$. By caching the values of the computed GP (posterior) marginal likelihoods, we substantially save on computational cost when computing expectations of the form $\mathbb{E}_{G \mid \boldsymbol{z}} \left[ p(\boldsymbol{x}^{1:t} \mid G)\, \phi(G) \right]$ and $\mathbb{E}_{G \mid \boldsymbol{z}} \left[ p(\boldsymbol{x}^{t+1} \mid G, \boldsymbol{x}^{1:t})\, \phi(G) \right]$ where $\phi(G)$ is some quantity depending the graph.

Specifically, consider that $p(\mathbf{x}^{1:t}|G) = \prod_i p(\mathbf{x}_i^{1:t}|G)$ factorizes into the GP marginal likelihoods of the individual mechanisms, so for $d$ nodes in the graph and $N$ samples in $\mathbf{x}^{1:t}$ (counted over all time steps) the complexity of computing $p(\mathbf{x}^{1:t}|G)$ is $O(d \cdot N^3)$ for a fixed set of GP hyperparameters (for

simplicity we ignore that not all $d$ mechanisms are modelled by GPs as some are root nodes, so this is not a tight bound). Thus, in the worst case, estimating $p(\mathbf{x}^{1:t}|z) = \mathbb{E}_{G|z}[p(\mathbf{x}^{1:t}|G)]$ with K graph samples would yield a complexity of $O(K \cdot d \cdot N^3)$. By caching the marginal likelihoods as outlined above we can rewrite the complexity $O(K \cdot d \cdot N^3)$ as $O(L \cdot N^3)$ where $L \leq K \cdot d$ denotes the number of unique mechanisms entailed by the set of $K$ graph samples. Although this does not reduce the worst case complexity it nevertheless greatly alleviates the computational demand in practice.

The benefit of caching becomes even more pronounced as $p(G \,|\, \boldsymbol{z})$ concentrates is mass on a small set of similar graphs as a result of the inference process. In particular, when updating the latent particles using SVGD we do not need to recompute $p(\boldsymbol{x}^{1:t} \,|\, G)$ after we have once before sampled $G$, which greatly speeds up the gradient estimation of the particle posterior.

### E.3    Computational Complexity

There are two main phases in our algorithm (disregarding the computation of metrics for evaluation), (i) the inference phase where we (approximately) infer the posterior over SCMs $p(\mathcal{M}|\mathbf{x}^{1:t})$ after collecting new experimental data, and (ii) the experimental design phase.

The inference phase has worst-case complexity in $O(T_{SVGD} \cdot (T_{HP} \cdot M \cdot K \cdot d \cdot N^3 + M^2 \cdot d^2))$ where $T_{SVGD}$ is the number of SVGD update steps, $T_{HP}$ is the number of GP hyperparameter update steps, $M$ is the number of latent $z$ particles, $K$ is the number of graph samples per latent $z$ particle, $d$ is the number of nodes in the network, and $N$ is the number of collected experimental samples in $\mathbf{x}^{1:t}$. The computation of the GP marginal likelihood dominates the complexity of the inference phase. To improve scalability we make use of shared priors and caching. Additionally, we update the GP hyperparameters according to a predefined schedule instead of doing so after each performed experiment. In our experiments, both measures reduce the factor $T_{HP} \cdot M \cdot K \cdot d$ significantly. For example, running inference with 5 freshly initialized $z$ particles with 40 graph samples each on a scale-free SCM with 20 nodes updates the hyperparameters of (2970, 964, 177) GPs during SVGD update steps (1, 5, 10), and of less than 15 GPs after 20 SVGD update steps. Compared to $M \cdot K \cdot d = 4000$ in this example, the benefit is evident.

In the experimental design phase we parallelize finding the optimal intervention value for each candidate target node, so the complexity is basically the number of Bayesian optimization (BO) steps times the complexity of the utility we want to optimize for. For a general query (cf. Eq. (4.11)) we have complexity in $O(T_{BO} \cdot M \cdot K_{outer} \cdot S \cdot Q \cdot M \cdot K_{inner} \cdot (O(p(y|\mathcal{M}) + d \cdot N^3)))$ where $T_{BO}$ is the number of Bayesian optimization iterations, $M$ is the number of latent $z$ particles, $K_{outer}$ is the number of graph samples in the outer SCM expectation in, $S$ is the number of simulated experiments per SCM, $Q$ is the number of simulated queries, $K_{inner}$ is the number of graph samples in the inner SCM expectation, $O(\,p(y|\mathcal{M})\,)$ is the complexity of evaluating the query likelihood and $d \cdot N^3$ is the complexity of evaluating the GP predictive posteriors for the simulated experiments. For the causal discovery and model learning utilities the complexity reduces to $O(T_{BO} \cdot M \cdot K_{outer} \cdot S \cdot M \cdot K_{inner} \cdot d \cdot N^3)$.

In summary, the complexity of our ABCI implementation is dominated the experimental design phase from a high-level perspective. On a lower level, the cubic scaling of GP inference is the major computational issue that we alleviate by caching the (posterior) marginal log-likelihoods (see Appx. E.2 for details). However, in a small data regime where experimental data is costly to obtain, GPs are not a prohibitive element in our inference chain. Furthermore, GP scaling issues could be alleviated, e.g., by using sparse GP approximation or any other kind of scalable Bayesian mechanism model. Disregarding issues of GP scaling, the estimation of the information gain utilities is still costly, simply because it requires many levels of nested sampling and too few Monte-Carlo samples will yield too noisy, in the worst case unusable utility estimates. We believe that in follow-up work much can be gained in terms of scalability as well as performance by incorporating recent advances in nested Monte-Carlo/information gain estimation techniques(e.g., [8, 28, 63]).

Finally, consider that a single estimation of the causal discovery utility for an SCM with 20 nodes with $N = 500$ previously collected experimental samples takes approximately 2 minutes on an off-the-shelf laptop. Thus, for 10 BO iterations we can do the experimental design phase in 20 minutes (assuming we parallelize the utility optimization for each node). In a practical application scenario one might be very willing to invest hours or days for the design phase before conducting a costly experiment.

### E.4 Implementation and Computing Resources

Our Python implementation uses the PyTorch [54], GPyTorch [24], CDT [41], SKLearn [58], NetworkX [29] and BoTorch [5] packages, which greatly eased our implementation efforts. All of our experiments were run on CPUs. We parallelise the experiment design by running the optimisation process for each candidate intervention set on a separate core.

# F Evaluation Metrics

In this section, we provide details on the metrics used to evaluate our method in Section 5 and Appx. G. In our experiments, we use (nested) Monte Carlo estimators to approximate intractable expectations.

**Kullback-Leibler Divergence.** We evaluate the inferred posterior over queries given observed data, $p(Y \mid \boldsymbol{x}^{1:t})$, to the true query distribution $p(Y \mid \mathcal{M}^\star)$ using the Kullback-Leibler Divergence (KLD), i.e.,

$$\mathrm{KL}(p(Y \mid \mathcal{M}^\star) \| p(Y \mid \boldsymbol{x}^{1:t})) = \mathbb{E}_{Y \mid \mathcal{M}^\star} \left[ \log p(Y \mid \mathcal{M}^\star) - \log p(Y \mid \boldsymbol{x}^{1:t}) \right] \tag{F.1}$$

$$= \mathbb{E}_{Y \mid \mathcal{M}^\star} \left[ \log p(Y \mid \mathcal{M}^\star) - \log \mathbb{E}_{\mathcal{M} \mid \boldsymbol{x}^{1:t}} [p(Y \mid \mathcal{M})] \right]. \tag{F.2}$$

**Query KLD.** For $Y = X_5^{\mathrm{do}(X_3=\psi)}$ with $\psi \sim p(\psi)$ we have

$$\text{Query KLD} = \mathbb{E}_\psi \left[ \mathrm{KL}(p^{\mathrm{do}(X_3=\psi)}(X_5 \mid \mathcal{M}^\star) \| p^{\mathrm{do}(X_3=\psi)}(X_5 \mid \boldsymbol{x}^{1:t})) \right] \tag{F.3}$$

$$= \mathbb{E}_\psi \left[ \mathbb{E}_{X_5 \mid \mathrm{do}(X_3=\psi), \mathcal{M}^\star} \left[ \log p^{\mathrm{do}(X_3=\psi)}(X_5 \mid \mathcal{M}^\star) - \log p^{\mathrm{do}(X_3=\psi)}(X_5 \mid \boldsymbol{x}^{1:t}) \right] \right]. \tag{F.4}$$

To approximate the outer two expectations, we keep a fixed set of samples for each ground truth SCM to enhance comparability between different ABCI runs. For $p^{\mathrm{do}(X_3=\psi)}(X_5 \mid \boldsymbol{x}^{1:t})$, we use the estimator described in Section D.1.

**SCM KLD.** For $Y = q_{\mathrm{CML}}(\mathcal{M}) = \mathcal{M}$, we have

$$\text{SCM KLD} = \mathrm{KL}(p(\mathcal{M} \mid \mathcal{M}^\star) \| p(\mathcal{M} \mid \boldsymbol{x}^{1:t})) \tag{F.5}$$

$$= \mathbb{E}_{\mathcal{M} \mid \mathcal{M}^\star} \left[ \log p(\mathcal{M} \mid \mathcal{M}^\star) - \log p(\mathcal{M} \mid \boldsymbol{x}^{1:t}) \right] \tag{F.6}$$

$$= 0 - \log p(\mathcal{M}^\star \mid \boldsymbol{x}^{1:t}) \tag{F.7}$$

$$= -\log \mathbb{E}_{\mathcal{M} \mid \boldsymbol{x}^{1:t}} [p(\mathcal{M}^\star \mid \mathcal{M})] \tag{F.8}$$

$$= -\log \mathbb{E}_{G, \boldsymbol{f}, \boldsymbol{\sigma}^2 \mid \boldsymbol{x}^{1:t}} \left[ p(G^\star, \boldsymbol{f}^\star, \boldsymbol{\sigma}^{2,\star} \mid G, \boldsymbol{f}, \boldsymbol{\sigma}^2) \right] \tag{F.9}$$

$$= -\log \mathbb{E}_{G, \boldsymbol{f}, \boldsymbol{\sigma}^2 \mid \boldsymbol{x}^{1:t}} \left[ p(G^\star \mid G) \, p(\boldsymbol{f}^\star \mid \boldsymbol{f}) \, p(\boldsymbol{\sigma}^{2,\star} \mid \boldsymbol{\sigma}^2) \right]. \tag{F.10}$$

Now note that $p(G^\star \mid G) = 1$ if $G = G^\star$ and 0 otherwise, $p(\boldsymbol{f}^\star \mid \boldsymbol{f}) = \delta(\boldsymbol{f}^\star - \boldsymbol{f})$, and $p(\boldsymbol{\sigma}^{2,\star} \mid \boldsymbol{\sigma}^2) = \delta(\boldsymbol{\sigma}^{2,\star} - \boldsymbol{\sigma}^2)$. Hence, the SCM KLD vanishes iff the SCM posterior $p(\mathcal{M} \mid \boldsymbol{x}^{1:t})$ collapses onto the true SCM $\mathcal{M}^\star$, and is infinite otherwise.

**Average Interventional KLD.** Computing the KLD for $Y = q_{\mathrm{CML}}(\mathcal{M}) = \mathcal{M}$ is not useful for evaluation, since it vanishes when the SCM posterior $p(\mathcal{M} \mid \boldsymbol{x}^{1:t})$ collapses onto the true SCM $\mathcal{M}^\star$ and is infinite otherwise. For this reason, we report the *average interventional KLD* as a proxy metric, which we define as

$$\text{Avg. I-KLD} = \frac{1}{d} \sum_{i=1}^d \mathbb{E}_\psi \left[ \mathrm{KL}(p^{\mathrm{do}(X_i=\psi)}(X \mid \mathcal{M}^\star) \| p^{\mathrm{do}(X_i=\psi)}(X \mid \boldsymbol{x}^{1:t})) \right] \tag{F.11}$$

$$= \frac{1}{d} \sum_{i=1}^d \mathbb{E}_\psi \left[ \mathbb{E}_{X \mid \mathrm{do}(X_i=\psi), \mathcal{M}^\star} \left[ \log p^{\mathrm{do}(X_i=\psi)}(X \mid \mathcal{M}^\star) - \log p^{\mathrm{do}(X_i=\psi)}(X \mid \boldsymbol{x}^{1:t}) \right] \right] \tag{F.12}$$

$$= \frac{1}{d} \sum_{i=1}^d \mathbb{E}_\psi \left[ \mathbb{E}_{X \mid \mathrm{do}(X_i=\psi), \mathcal{M}^\star} \left[ \log p^{\mathrm{do}(X_i=\psi)}(X \mid \mathcal{M}^\star) \right. \right. \tag{F.13}$$

$$\left. \left. - \log \mathbb{E}_{\mathcal{M} \mid \boldsymbol{x}^{1:t}} \left[ p^{\mathrm{do}(X_i=\psi)}(X \mid \mathcal{M}) \right] \right] \right].$$

As with the Query KLD, we keep a fixed set of MC samples per ground truth SCM to approximate the two outer expectations to enhance comparability between different ABCI runs.

**Expected Structural Hamming Distance.** The Structural Hamming Distance (SHD)

$$\mathrm{SHD}(G, G^\star) = \left| \{ (i,j) \in G : (i,j) \notin G^\star \} \right| + \left| \{ (i,j) \in G^\star : (i,j) \notin G \} \right| \tag{F.14}$$

denotes the simple graph edit distance, i.e., it counts the number of edges $(i,j)$ that are present in the prediction graph $G$ and not present in the reference graph $G^\star$ and vice versa. We report the expected SHD w.r.t. our posterior over graphs as

$$\text{ESHD}(G, G^\star) = \mathbb{E}_{G \,|\, \boldsymbol{x}^{1:t}} \left[ \text{SHD}(G, G^\star) \right] \tag{F.15}$$

**AUPRC.** Following previous work [19, 21, 45, 78], we report the *area under the precision recall curve* (AUPRC) by casting graph learning as a binary edge prediction problem given our inferred posterior edge probabilities $p(G_{i,j} \,|\, \boldsymbol{x}^{1:t})$. Refer to e.g. Murphy [52] for further information on this quantity.

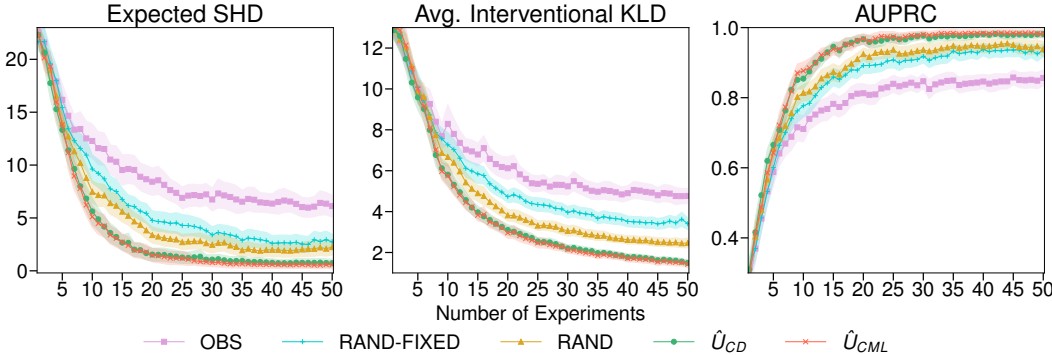

Figure 5: **Causal Discovery and SCM Learning on Scale-free Graphs with** 10 **Variables.** Comparison of the experimental design strategies with random and observational baselines on simulated ground truth models with 10 nodes. Lines and shaded areas show means and 95% confidence intervals (CIs) across 50 runs (10 randomly sampled ground-truth SCMs with 5 restarts per SCM). The $U_{\text{CD}}$ and $U_{\text{CML}}$ objectives perform on par with each other. Both clearly outperform the observational and random baselines on all metrics.

# G    Extended Experimental Results

**Causal Discovery and SCM Learning for SCMs with** $d = 10$ **Variables.**    We report results on ground truth SCMs with $d = 10$ variables and scale-free graphs in Fig. 5. We initialise all methods with 5 observational samples and perform experiments with a batch size of 3. All other parameters are chosen as described in Appx. D.

**Causal Discovery and SCM Learning for SCMs with** $d = 20$ **Variables.**    To demonstrate the scalability of our framework, we report results on ground truth SCMs with $d = 20$ variables and scale-free or Erdős-Renyi graphs in Fig. 6 and Fig. 7, respectively. We initialise all methods with 50 observational samples and perform experiments with a batch size of 5. All other parameters are chosen as described in Appx. D.

While ABCI shows clear benefits when scale-free causal graphs underlie the SCMs, we find that the advantage of ABCI diminishes on SCMs with unstructured Erdős-Renyi graphs, which appear to pose a harder graph identification problem. Moreover, we expect performance of our inference machinery, especially together with the informed action selection, to increase when investing more computational power to improve the quality of our estimates, e.g., by increasing the number of Monte Carlo samples used in our estimators and increasing the number of evaluations during the Bayesian optimisation phase.

Finally, in Fig. 8 we show that using a simple linear model (GP model with a linear kernel) is not able to reasonably capture the characteristics of the ground truth model (non-linear GP model) due to the model mismatch.

**Learning Interventional Distributions vs. Causal Discovery and SCM Learning.**    We report additional metrics for our causal reasoning experiment as described in § 5 in Figs. 9 and 10. The key result here is that $U_{\text{CR}}$ yields a significantly lower Query KLD while exhibiting a worse ESHD and Average I-KLD scores, which indicates that, indeed, the $U_{\text{CR}}$ learns only those parts of the model that are relevant to reducing the uncertainty in our target query. This is more data efficient than trying to learn the entire model first and then answering the causal query of interest.

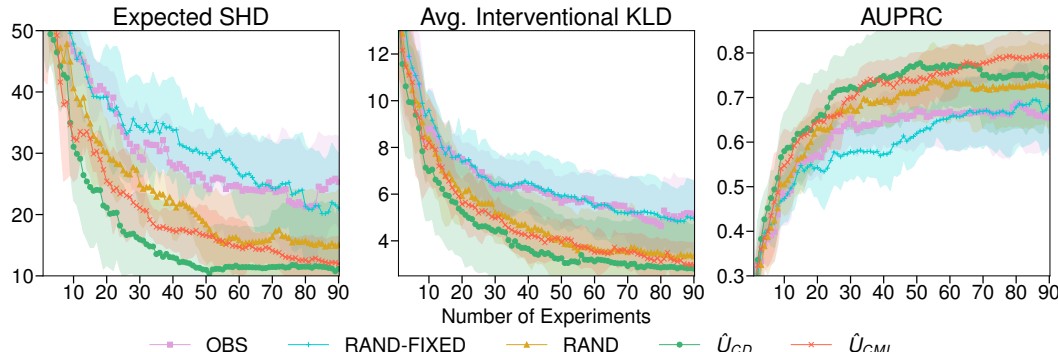

Figure 6: **Causal Discovery and SCM Learning on Scale-free Graphs with** 20 **Variables.** (Same figure as in Fig. 3 with additional confidence intervals for OBS and RAND FIXED.) Comparison of the experimental design strategy for causal discovery ($U_{\mathrm{CD}}$) with random and observational baselines on simulated ground truth models with 20 nodes. Lines and shaded areas show means and 95% confidence intervals (CIs) across 15 runs (5 randomly sampled ground-truth SCMs with 3 restarts per SCM). The $U_{\mathrm{CD}}$ objective significantly outperforms the observational and random baselines on all metrics.

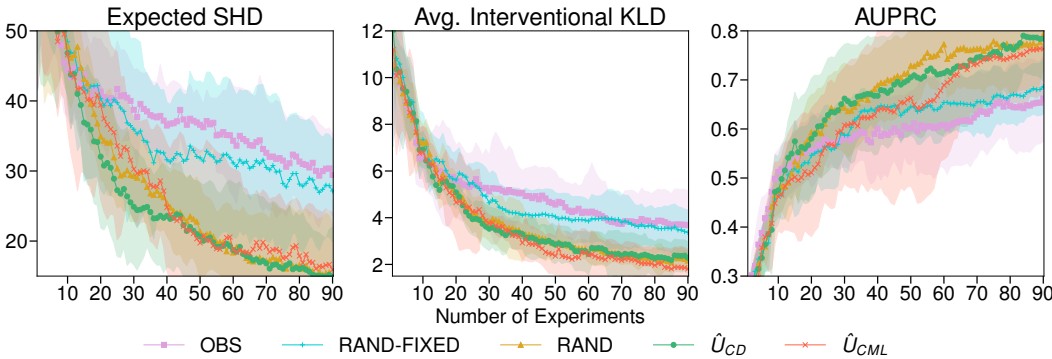

Figure 7: **Causal Discovery and SCM Learning on Erdős-Renyi Graphs with** 20 **Variables.** Comparison of experimental design strategies for causal discovery ($U_{\mathrm{CD}}$) and causal model learning ($U_{\mathrm{CML}}$) with random and observational baselines on simulated ground truth models with 20 nodes. Lines and shaded areas show means and 95% confidence intervals (CIs) across 15 runs (5 randomly sampled ground-truth SCMs with 3 restarts per SCM). The $U_{\mathrm{CD}}$ and $U_{\mathrm{CML}}$ strategies perform approx. equal to the strong random baseline (RAND) on all metrics, however, all three are significantly better than the weak random (RAND FIXED) and observational baselines. We expect that improving the quality of the $U_{\mathrm{CD}}$ and $U_{\mathrm{CML}}$ estimates (e.g., by scaling up computational resources invested in the MC estimates) yield similar benefits of the experimental design utilities as apparent in Fig. 6.

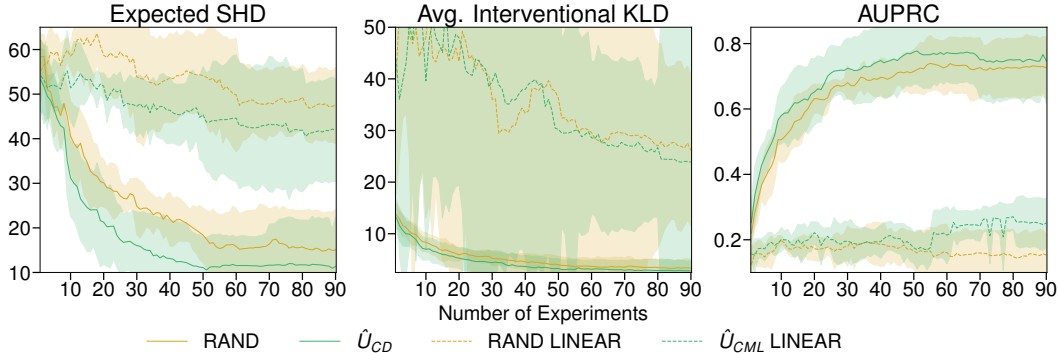

Figure 8: **Causal Discovery and SCM Learning on Scale-free Graphs with** 20 **Variables.** Comparison of non-linear GP model with a linear model (linear GP kernel) for $U_{\mathrm{CD}}$ an RAND on simulated ground truth models with 20 nodes. Lines and shaded areas show means and 95% confidence intervals (CIs) across 15 runs (5 randomly sampled ground-truth SCMs with 3 restarts per SCM). Clearly, the model mismatch in the linear model prohibits the identification of the ground-truth graph.

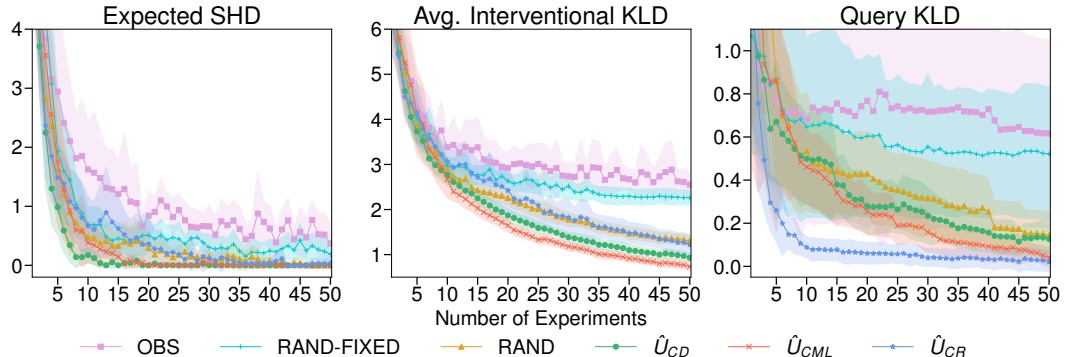

Figure 9: **Learning Interventional Distributions.** Comparison of the experimental design strategies with random and observational baselines. Lines and shaded areas show means and 95% confidence intervals (CIs) across 30 runs (10 randomly sampled ground-truth SCMs with 3 restarts per SCM). $U_{CD}$, $U_{CML}$ and $U_{CR}$ perform best w.r.t. the ESHD, Avg. I-KLD and Query KLD metrics respectively, which is expected.

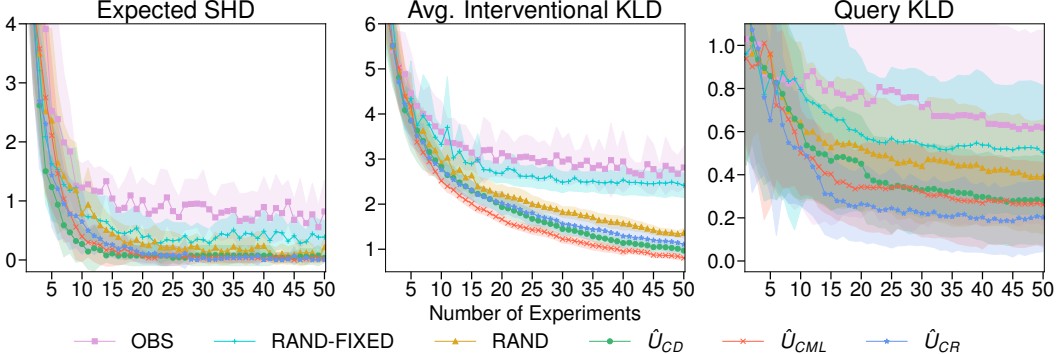

Figure 10: **Learning Interventional Distributions.** Comparison of the experimental design strategies with random and observational baselines. Lines and shaded areas show means and 95% confidence intervals (CIs) across 30 runs (10 randomly sampled ground-truth SCMs with 3 restarts per SCM). $U_{CD}$, $U_{CML}$ and $U_{CR}$ perform best w.r.t. the ESHD, Avg. I-KLD and Query KLD metrics respectively, which is expected.