# OpenReview forum: "Active Bayesian Causal Inference"
_NeurIPS.cc/2022/Conference — NeurIPS 2022 Accept_

### Official Review · Reviewer_T1JL · 2022-07-10

**Rating:** 6
**Confidence:** 4
**Soundness:** 3 good
**Presentation:** 4 excellent
**Contribution:** 3 good

**Summary:**

This work proposes Active Bayesian Causal Inference (ABCI) that is a fully-Bayesian active learning framework for integrated causal discovery and reasoning, which jointly infers a posterior over causal models and queries of interest. They implemented ABCI for the class of nonlinear additive Gaussian noise models using Gaussian processes. To parametrize the combinatorial space of causal graphs and having tractable posterior, they applied a continuous latent probabilistic graph representation.

**Questions:**

Please see the above comments.

**Limitations:**

Please see the above comments.

**Strengths And Weaknesses:**

They studied a practical and important question that is answering to a causal query without knowing the causal graph. The paper is well-written.

The proposed approach, unlike most exiting methods, does not learn the causal structure and then answer to the causal query of interest. This makes it both novel and more data efficient.

Although the proposed ABCI is a general framework but as it is mentioned it is not a trackable algorithm for general causal models and hence it is implemented for nonlinear Gaussian models. How can this algorithm be used for model classes that contain discrete variables?

Applying information gain framework for selecting the next best intervention is standard in Bayesian approaches but it is suboptimal as it is greedy. Is there any guarantee of how far the solution is from the optimal solution (e.g., if the objective is sub-modular then it is a constant approximation)? Moreover, as it is suggested by the experiments, there is not so much gain compared to the random selection method considering the additional computational cost of design experiment line in ABCI algorithm.

The overall method is computationally intense which makes its scalability questionable. A discussion on the computational complexity of ABCI would be interesting. The experiments are also for small graphs.

As it is mentioned, this work is built on causal sufficiency which is a limiting assumption and in most real-world applications it does not hold. Having said this fact, suppose that the target causal query is structure learning. In this case, having observational data, we can learn the essential graph of the corresponding MEC. On the other hand, the authors of “Learning Causal Graphs with Small Interventions” show that designing minimum number of interventions with limited size to learn the causal structure is NP-hard. When the actable set of variables are limited in size, then it is interesting to see how ABCI algorithm performs compared to the aforementioned work.

Under the causal sufficiency and using the rules of do-calculus, all interventional queries become identifiable from observational data given the causal graph. It is important to see whether ABCI algorithm in the learning interventional distribution task learns the whole graph or only a part of it before finding the interventional distribution of interest.

Minor:
Introduction: "Additional structural assumptions (e.g., linearity)". Linearity is not a structural assumption. It is an assumption on the underlying generative model.

---

> ### Author Response · Authors · 2022-08-02
> **Response to Reviewer T1JL - Part 2**
>
> ---
>
> > As it is mentioned, this work is built on causal sufficiency which is a limiting assumption and in most real-world applications it does not hold. Having said this fact, suppose that the target causal query is structure learning. In this case, having observational data, we can learn the essential graph of the corresponding MEC. On the other hand, the authors of “Learning Causal Graphs with Small Interventions” show that designing minimum number of interventions with limited size to learn the causal structure is NP-hard. When the actable set of variables are limited in size, then it is interesting to see how ABCI algorithm performs compared to the aforementioned work.
>
> Regarding causal sufficiency, as we point out in the discussion, our general ABCI framework does allow for hidden confounding, but our practical implementation does not as this is non-trivial in terms of modeling and tractable inference.
>
> Thank you for pointing us to this work [1]. While certainly related, we believe that there are some important differences in the problem setup between [1] and our paper. Specifically, [1] assumes that interventions are perfectly informative about the underlying graph structure, i.e., that all conditional independences induced by the underlying intervened graph can be inferred from the experimental data with certainty. Hence, in [1] once a certain intervention is performed, the induced interventional-MEC (I-MEC) is obtained  and *performing the same experiment again will not yield any additional information.*
>
> In contrast, in our problem setting we assume that only finitely many data points are available (observational and per experiment/intervention). This induces a substantial uncertainty in the conclusions we can draw about the underlying graph. In other words, when we perform an intervention we cannot infer the (I-MEC) with certainty and hence, we may want to *perform this intervention again*, as it will still yield information about the underlying graph.
>
> So, whereas [1] tries to minimize the number of unique interventions with unlimited data per intervention to fully identify the true graph, without considering  uncertainty in their inferences, we try to perform at each time step (possibly repeating) interventions to maximally reduce our uncertainty in the true graph. As another point of difference, we also optimize for the actual intervention value, whereas [1] only optimizes for the intervention targets.
> Due to these differences, we think that a fair empirical comparison of both works is hard and an implementation of [1] for finite data is not available to the best of our knowledge.
>
> We added this discussion to our section on further related work in Appendix A.
>
> [1] Shanmugam, K., Kocaoglu, M., Dimakis, A. G. & Vishwanath, S. Learning Causal Graphs with Small Interventions. in *Advances in Neural Information Processing Systems* (eds. Cortes, C., Lawrence, N., Lee, D., Sugiyama, M. & Garnett, R.) vol. 28 (Curran Associates, Inc., 2015).
>
> ---
>
> >Under the causal sufficiency and using the rules of do-calculus, all interventional queries become identifiable from observational data given the causal graph. It is important to see whether ABCI algorithm in the learning interventional distribution task learns the whole graph or only a part of it before finding the interventional distribution of interest.
>
> Although (interventional) queries may be identifiable from observational data, having only finitely many data will nevertheless induce uncertainty about the queries. Naturally, in our finite/small data regime, we will have uncertainty in both, the graph and the query, at the same time. What we see see in Figs. 9 and 10 in Appendix E in the revised document is, that for the $U_{CR}$ utility (learning interventional distributions) we have lower uncertainty in the query while having a higher uncertainty in the graph when compared to the $U_{CD}$ (learning the graph) and $U_{CML}$ (learning the SCM) utilities. This empirically shows what is expected from a theoretical Bayesian perspective: our utilities yield experiments that are maximally informative about our target query whereas uncertainty about other aspects of the underlying SCM is not considered/marginalised out in this process.
>
> ---
>
> >Minor: Introduction: "Additional structural assumptions (e.g., linearity)". Linearity is not a structural assumption. It is an assumption on the underlying generative model.
>
> Thank you for pointing this out. What we meant are assumptions on the form of the structural equations, but we see that this can be seen as ambiguous. We changed the phrasing to "Additional assumptions (e.g., linearity) [...]".

---

> ### Author Response · Authors · 2022-08-02
> **Response to Reviewer T1JL - Part 1**
>
> ---
>
> > Although the proposed ABCI is a general framework but as it is mentioned it is not a trackable algorithm for general causal models and hence it is implemented for nonlinear Gaussian models. How can this algorithm be used for model classes that contain discrete variables?
>
> Indeed, ABCI is a general framework and the concrete modeling choices and (approximate) inference techniques used will depend on the problem at hand. One of our contributions was to provide an implementation for continuous variables with nonlinear relationships, for which no such method was previously available.
> For discrete variables, exact inference through a conjugate Dirichlet-Multinomial model is actually straight forward, and was previously considered, e.g., in the cited work of Murphy (2001) and Tong and Koller (2001), see also [1].
>
> In our response to reviewer `8rNg` we also write "the methods chosen in our implementation can certainly be exchanged depending on the problem setup at hand, as long as we can estimate posterior expectations over SCMs given collected data $\mathbf{x}^{1:t}$ of the form $\mathbb{E}_{\mathcal{M}| \mathbf{x}^{1:t}}[\cdot]$".
>
> [1] David Heckerman, Dan Geiger, and David M. Chickering. Learning Bayesian Networks: The Combination of Knowledge and Statistical Data. Machine Learning, 20(3):197–243, September 1995.
>
> ---
>
> > Applying information gain framework for selecting the next best intervention is standard in Bayesian approaches but it is suboptimal as it is greedy. Is there any guarantee of how far the solution is from the optimal solution (e.g., if the objective is sub-modular then it is a constant approximation)? Moreover, as it is suggested by the experiments, there is not so much gain compared to the random selection method considering the additional computational cost of design experiment line in ABCI algorithm.
>
> To discuss sub-optimality, we distinguish between two scenarios: (i) we have budget constraints (e.g., a fixed number of samples to be allocated to a fixed number of experiments as in [1]) and are interested in the inference results only after we have exhausted our budget, vs. (ii) we have no such constraints and assume that our inference results are used after every experiment cycle.
>
> In this work, we assume the latter scenario (ii) and  argue that in this case the information gain objective is not sub-optimal. Nevertheless, we agree that the former scenario is an interesting and practically relevant one. The authors of earlier related work on linear mechanisms [1] show that the information gain is sub-modular for the target query $q(\mathcal{M}) = G$, i.e., for causal discovery, and they conjecture that this also holds for general functions of the graph (cf. [1, Thm. 4.1]). However, we are not aware of any follow-up results on this. Although sub-modularity of our general query utility would be a strong additional result, we leave this for future work.
>
> Regarding computational cost vs. the performance gap between experimental design (ED) and random strategy, we have updated the main paper with new experimental results showing that (i) the gap increases for the causal discovery and causal model learning tasks on scale-free graphs with 20 nodes, and (ii) the gap increased for the causal reasoning task after we identified and corrected an error in the implementation of the utility estimation.
> From a practical viewpoint, the cost of conducting ED can only be justified when outweighed by the cost expected from performing random experiments. This depends on (i) how costly it is to perform an experiment, and (ii) how large the gap between ED and random selection (in terms of information gain) is. Both depend on the problem instance at hand and cannot be answered in general, so we leave this question to be answered by the user.
>
> [1] Agrawal, R., Squires, C., Yang, K., Shanmugam, K. & Uhler, C. ABCD-Strategy: Budgeted Experimental Design for Targeted Causal Structure Discovery. in *Proceedings of Machine Learning Research* (eds. Chaudhuri, K. & Sugiyama, M.) vol. 89 3400–3409 (PMLR, 2019).
>
> ---
>
> > The overall method is computationally intense which makes its scalability questionable. A discussion on the computational complexity of ABCI would be interesting. The experiments are also for small graphs.
>
> We refer you to our corresponding answer to reviewer `LPpv`. As already mentioned above, we have updated results for SCMs with 20 nodes to show our scalability to larger problem instances. Additionally, we included a detailed discussion on the computational complexity of our approach in a novel section in Appendix D.10.

---

### Official Review · Reviewer_LPpv · 2022-07-10

**Rating:** 9
**Confidence:** 5
**Soundness:** 4 excellent
**Presentation:** 4 excellent
**Contribution:** 4 excellent

**Summary:**

This paper introduces a Bayesian approach to causality that combines causal discovery (of the causal graph and mechanisms) and inference (about a question of interest). They use GPs with additive noise for the mechanisms to be able to have analytics posteriors for them. They use the previously introduced DiBS framework to optimize over a soft version of the graph (using a continuous latent variable Z that conditions independent edge choices) using a variational method. They propose a generic method for handling different kinds of causal inference queries (e.g. to discover the graph vs to discover a particular edge or estimate a causal effect on some variable given some hypothetical interventino). An active learning Bayesian experimental design algorithm is proposed which exploits the estimated posteriors to estimate and optimize information gains wrt possible interventions.

**Questions:**

1. first sentence after eq 3.3 may be confusing for some readers, since Y may be a function of some condition or intervention; what the authors probably mean is that q(M) is a posterior that concentrates mass on a single distribution (whereas 3.3 is a mixture and P(Y|M) is not a Dirac).
2. line 203: why do we need to distinguish between root nodes and non-root nodes? (see also next 2 questions)
3. Eq 4.4: why can we factor p(f,sigma|G) for NR but not for R?
4. line 212: why is this true for NR but not for R?
5. sec 4.1: do we need to solve (i.e. O(n^2) time) for the GPs of all the nodes after each sample of graph G? is that the computational bottleneck?
6. how does the overall algorithm complexity scale with the size of the graph and the number of examples? what are the computational bottlenecks in the overall procedure?


**Limitations:**

- see the above questions about computational complexity, which should be discussed in the limitations section too


**Strengths And Weaknesses:**

Pros:
- The approach is very appealing because it is highly principled, based on a Bayesian framework that is important to
- It is original in addressing together the causal discovery and the causal inference questions, something very important in because in general (especially for finite data and the generally unidentifable settings) there will be many more than one graph G that is consistent with the given data.

Cons:
- the additive noise and GP assumptions are convenient but it would be good to think about getting rid of them (although this is clearly for future work, this paper is already quite dense)
- the computational complexity of the proposed approach should be clarified (see below): the fact that experiments are done with d=8 variables and starting with n=5 examples and minibatches of size 3 suggests that computational scaling may be an issue that should be better covered in the limitations section.

---

> ### Author Response · Authors · 2022-08-02
> **Response to Reviewer LPpv**
>
> > first sentence after eq 3.3 may be confusing for some readers, since Y may be a function of some condition or intervention; what the authors probably mean is that q(M) is a posterior that concentrates mass on a single distribution (whereas 3.3 is a mixture and P(Y|M) is not a Dirac).
>
> This notation may indeed be confusing. What is meant is that $q$ is a 'query' function, which returns the desired quantity of the causal model $\mathcal{M}$. $P(Y|\mathcal{M})$ is meant to be a point mass on $q(\mathcal{M})$. Furthermore, it is correct that $Y = q(\mathcal{M})$ may depend on additional inputs (value of conditioning variable, an intervention, or values of exogenous variables). In our manuscript, we implied that the return value of $q$ might be a function (or density) -- this is a rather mathematical description and does not reflect the actual implementation, where we represent such function/density via the SCMs contained in the (approximate) posterior. We will clarify this point in the revision.
>
> ---
>
> > line 203: why do we need to distinguish between root nodes and non-root nodes? (see also next 2 questions)
>
> We need not necessarily distinguish between root and non-root nodes. Rather, this is a consequence of our modelling choices, for which we distinguish different priors and posterior inference techniques for root and non-root nodes. (E.g., GPs for modelling nonlinear dependence on the parents are only used for non-root nodes.)
> We modified the corresponding sentence accordingly.
>
> ---
>
> > Eq 4.4: why can we factor $p(f,\sigma|G)$ for NR but not for R?
>
> In short, whether $p(f_i, \sigma_i^2|G)$ factorizes or not is a consequence of our modeling choices to ensure tractable inference:
>
>  - the conjugate Normal-Inverse-Gamma prior on the mean and variance of root nodes entails dependent $f_i$ and $\sigma_i^2$,
>
>  - the assumption of independent priors over noise variances $\sigma_j^2$ and GP kernel parameters $\kappa_j$ entails the factorization $p(f_j, \sigma_j^2|G) = p(f_j|G) \cdot p(\sigma_j^2|G)$.
>
>
> ---
>
> > line 212: why is this true for NR but not for R?
>
> In fact, this also holds for root nodes as a consequence of the the causal sufficiency assumption. Thank you for catching this point; we adapted this in the revised paper.
>
> ---
>
> > sec 4.1: do we need to solve (i.e. O(n^2) time) for the GPs of all the nodes after each sample of graph G? is that the computational bottleneck?
> >
> > how does the overall algorithm complexity scale with the size of the graph and the number of examples? what are the computational bottlenecks in the overall procedure?
>
>
> There are two main phases in our algorithm, (i) the inference phase where we (approximately) infer the posterior over SCMs $p(\mathcal{M} | \mathbf{x}^{1:t})$ after collecting new experimental data, and (ii) the experimental design phase. The latter dominates the complexity of our overall procedure. For a general query (cf. Eq. (4.11)) we  get complexity in $O(T_{BO} \cdot M \cdot K_{outer} \cdot S \cdot Q \cdot M \cdot K_{inner} \cdot (O(p(y|\mathcal{M}) + d \cdot N^3)))$  where $T_{BO}$ is the number of Bayesian optimization iterations, $M$ is the number of latent $z$ particles, $K_{outer}$ is the number of graph samples in the outer SCM expectation in, $S$ is the number of simulated experiments per SCM, $Q$ is the number of simulated queries, $K_{inner}$ is the number of graph samples in the inner SCM expectation, $O(\,p(y|\mathcal{M})\,)$ is the complexity of evaluating the query likelihood and $d\cdot N^3$ is the complexity of evaluating the GP predictive posteriors for the simulated experiments.
>
> Hence, the asymptotic complexity of our ABCI implementation is dominated by the GP inference. Disregarding issues of GP scaling, the estimation of the information gain utilities is still costly, simply because it requires many levels of nested sampling and too few Monte-Carlo samples will yield too noisy, in the worst case unusable utility estimates.
>
> We alleviate some of the GP scaling issues by exploiting the shared priors assumption and using caching as discussed in Appendix D.6. We included a discussion of computational complexity, including the inference phase and the causal discovery and model learning utilities, in a new Appendix D.11.

---

### Official Review · Reviewer_8rNg · 2022-07-12

**Rating:** 6
**Confidence:** 4
**Soundness:** 3 good
**Presentation:** 3 good
**Contribution:** 3 good

**Summary:**

The paper introduces a new framework, ABCI, for end-to-end Bayesian causal inference. Starting from no data, ABCI iteratively designs experiments (interventions) to maximize the information gain for the target causal query. ABCI maintains uncertainty over the causal model over time by maintaining a set of latent particles, corresponding to causal graphs, which are used to compute the information gain (through a Gaussian-process based marginal likelihood). ABCI is distinguished from previous works by 1) targeting (and optimizing interventions) for a specific causal query $q(\mathcal{M})$, which can be any function of the SCM; and 2) modeling linear non-Gaussian SCMs. Empirically, ABCI methods are shown to provide improved performance compared to random-intervention baselines.


**Questions:**

Key questions:
- What are the essential novel insights/results that we should take away from this paper, that were not previously known?
- What is the justification (e.g. computational) for the choices made? For instance, can other models be used in place of Gaussian processes? Why was Bayesian optimization chosen? Can alternative Bayesian structure learning methods be used in place of DiBS?
- In Appx D.7, what likelihood $p(x|z)$ is used for SVGD, and why? Is it the GP marginal-likelihood?
- In Appx D.7, it is mentioned that only 5 latent particles are maintained while using DiBS; intuitively, this seems insufficient to maintain enough graphical uncertainty. Have the authors tested the effect of changing this hyperparameter?

Other:
- Is it possible to use gradient-based optimization for the utility (Eqn 4.11)? E.g. score function estimator?
- Have the authors considered using a Laplace approximation for the GP hyperparameter posterior?
- In line 268, it is shown that simplified utilities are available for cases where the query is either the full graph, or full SCM. Are there any practical benefits (e.g. in terms of estimation) in these cases?


**Limitations:**

The authors have appropriately addressed the limitations through discussion.

**Strengths And Weaknesses:**

This is a solid work which tackles the important problem of designing optimal experiments for answering causal queries, which are not identifiable from observational data. Bayesian inference is a principled approach through which to maintain uncertainty, and the ABCI framework is a sensible and well-engineered end-to-end solution for inferring a target causal query. In that sense, ABCI fills an practical gap that previous works did not fully address.

On the positive side, ABCI brings together a number of previous ideas, such as maintaining uncertainty through graph samples, maximizing information gain, and using Gaussian processes to model non-linear relationships with tractable marginal likelihoods, to form a useful end-to-end framework. The paper investigates targeting specific queries by developing utility functions (information gain) for arbitrary functions of the SCM. The benefits are demonstrated experimentally, where $U_{CR}$ is shown to converge faster than $U_{CML}$ on the specific causal query.

My key concern is that there does not appear to be significant novel components introduced in this work; nor does there appear to be significant novelty in the approach to integrating the components: for instance, computation appears to be the key limiting factor, and there does not appear to be theoretical advances in this regard. Alternatively, comprehensive empirical analysis could provide justification and insight regarding the efficiencies and bottlenecks in the ABCI pipeline, e.g. ablation studies considering different approaches to each step.

More technically, I have some concern over the nature of the uncertainty representation maintained by ABCI. In particular, ABCI maintains a set of latent particles $\textbf{Z}$, corresponding to graphs, and it appears that the GP marginal-likelihood and posterior is computed fresh in each timestep $t$. This is concerning for two reasons. Firstly, the number of samples maintained by DiBS (Appx D.7) is chosen to be very small (5), which seems unlikely to provide a sufficiently diverse uncertainty. Secondly, the idea of maintaining a set of graphs for evaluating information gain already appeared in Agrawal et al. [1], and the method proposed appears to just use this without taking advantage of modeling the full SCM and targeting a specific query (e.g. maintaining GP posterior also), which is the key novel component of this work.

From a presentation standpoint, the problem is well motivated and the prose is generally of very high-quality; I found the paper to be very pleasant to read. The description of Bayesian causal inference is thorough, perhaps the clearest that I have seen in the literature. However, in my view the paper goes too far in this direction; too much of the main paper is describing (the use of) existing work, without spending enough time highlighting the novel contributions, and perhaps even more importantly, justifying the choices being made (see Questions below). As a result, just reading the main paper, it’s difficult to tell what are the key novel insights.

Overall, I think the work is solid and a good practical contribution, but I am not entirely convinced that the novelty of the framework itself, or the analysis and insights thereof, is sufficiently strong.

---

> ### Author Response · Authors · 2022-08-02
> **Response to Reviewer 8rNg - Other Questions**
>
> ---
>
> > Is it possible to use gradient-based optimization for the utility (Eqn 4.11)? E.g. score function estimator?
>
>
> We decided to employ BO in our framework because of its conceptual simplicity and data efficiency.
> Gradient-based utility optimization would be an interesting avenue for future work, though we believe that the following three potential issues may require addressing:
>
> 1. The estimation of our utility functions is computationally demanding, so gradient-based optimization involving many update steps may become infeasible for larger problems.
>
> 2. Our utilities are in general non-convex, so gradient-based methods may suffer from poor local optima.
>
> 3. The score function estimator is known to yield very noisy gradient updates. As our utilities are estimated via nested Monte Carlo,  a very large number of samples may be required to obtain usable gradients.
>
>
>
> ---
>
> > Have the authors considered using a Laplace approximation for the GP hyperparameter posterior?
>
> This is a valid and interesting suggestion, but would also entail an increased computational effort in comparison to our current treatment, gradient-based MAP, which we empirically found sufficient for the current implementation.
>
>
>
> ---
>
> > In line 268, it is shown that simplified utilities are available for cases where the query is either the full graph, or full SCM. Are there any practical benefits (e.g. in terms of estimation) in these cases?
>
> Indeed, there are substantial benefits in terms of  computational complexity: compared to the general query utility, the causal discovery and model learning utilities do not require  simulating queries and evaluating their likelihoods. For more details, we refer to our response to reviewer `LPpv` below.

---

> ### Author Response · Authors · 2022-08-02
> **Response to Reviewer 8rNg - Key Questions**
>
> Before addressing your questions, we would like to briefly clarify/correct the following points from your summary.
>
> > "latent particles, corresponding to causal graphs"
>
> The latent particles $\mathbf{z}$ do not correspond to single graphs but encode a *distribution over graphs* $p(G | \mathbf{z})$.  Thus, $5$ latent particles encode a distribution over graphs that is more diverse than $5$ graph samples, which, as you correctly note, would be insufficient to capture complex or non-peaked posteriors.
>
>
> > "modeling linear non-Gaussian SCMs"
>
> We actually use GPs to model *non-linear* Gaussian SCMs (which is probably what was meant).
>
>
>
> We address your questions one by one below.
>
>
> ### Key Questions
>
> ---
>
> > What are the essential novel insights/results that we should take away from this paper, that were not previously known?
>
> Our main conceptual contribution is the integration of causal discovery and reasoning in a Bayesian end-to-end framework. Specifically, when compared to the common two-stage approach (discovery before reasoning), we treat uncertainty throughout the reasoning chain in a highly principled fashion. In contrast to previous works, e.g., the work of Agrawal et al. [S1], we do not only maintain a posterior over graphs *but a posterior over SCMs*, including graphs, mechanisms (posterior GPs), and noise parameters.
>
> On the practical side, we implement an instance of our theoretical ABCI framework with concrete assumptions on the underlying SCM to empirically demonstrate the effectiveness of our approach (see also the updated figures in our experiments section). We will share our implementation with the research community to facilitate further research and the extension of our approach.
>
> [S1] Agrawal, R., Squires, C., Yang, K., Shanmugam, K. & Uhler, C. ABCD-Strategy: Budgeted Experimental Design for Targeted Causal Structure Discovery. in *Proceedings of Machine Learning Research* (eds. Chaudhuri, K. & Sugiyama, M.) vol. 89 3400–3409 (PMLR, 2019).
>
> ---
>
> > What is the justification (e.g. computational) for the choices made? For instance, can other models be used in place of Gaussian processes? Why was Bayesian optimization chosen? Can alternative Bayesian structure learning methods be used in place of DiBS?
>
> ABCI is a general and modular theoretical framework, so it is true that the methods chosen in our implementation can be exchanged depending on the task, as long as we can estimate posterior expectations over SCMs given collected data $\mathbf{x}^{1:t}$ of the form $E_{\mathcal{M} | \mathbf{x}^{1:t}}[\cdot]$.
> For example, one could think of using (Bayesian) Neural Networks, Normalizing Flows or Linear Regression instead of GPs to accommodate different mechanisms and model classes.
> We chose Bayesian optimization (BO) instead of, e.g., gradient-based optimization methods, as the evaluation of our proposed utility functions is (i) costly and (ii) in general non-convex. In our evaluation, BO was not a significant computational factor, but simple search heuristics could be used as well.
> Finally, any (approximate) inference technique allowing us to estimate posterior expectations over acyclic graphs $\mathbb{E}_{G| \mathbf{x}^{1:t}}[\cdot]$ could be plugged into our framework instead of DiBS.
>
> ---
>
> > In Appx D.7, what likelihood p(x|z) is used for SVGD, and why? Is it the GP marginal-likelihood?
>
> As we use DiBS to approximately infer the posterior over graphs given collected data $p(G | \mathbf{x}^{1:t})$, the likelihood used for SVGD is consequently $$p(\mathbf{x}^{1:t} | \mathbf{z}) = \mathbb{E}_{G|\mathbf{z}}[p(\mathbf{x}^{1:t} | G)]$$ where estimating $p(\mathbf{x}^{1:t} | G)$ involves computing the GP marginal likelihoods of the mechanisms in $G$ and is detailed in Section 4.1.
>
> ---
>
> > In Appx D.7, it is mentioned that only 5 latent particles are maintained while using DiBS; intuitively, this seems insufficient to maintain enough graphical uncertainty. Have the authors tested the effect of changing this hyperparameter?
>
> As mentioned above, each latent $\mathbf{z}$ particle does not correspond to a single graph, but a *distribution over graphs* $p(G | \mathbf{z})$. When evaluating a posterior expectation over graphs, we sample a fixed number $K$ of graphs per latent particle. In our experiments, we used $M = 5$ latent particles with $K = 40$ graphs each, giving a total of $M \cdot K = 200$ graphs when estimating  $\mathbb{E}_{G| \mathbf{x}^{1:t}}[\cdot]$. In contrast to the original DiBS version, we do not use the annealing parameter $\alpha$ to force the mass of $p(G | \mathbf{z})$ onto a single graph during training. We clarified this in Appendix D.7 in our revised paper.
> We did not conduct a systematic evaluation on the effect of increasing the number of latent particles, since we found the approximation quality to be sufficient.

---

> > ### Comment · Reviewer_8rNg · 2022-08-08
> > **Thank you for the response**
> >
> > I would like to thank the authors for their comprehensive response to the reviews and changes made to the manuscript. Overall, I find that my concerns have been largely addressed, and have increased my score accordingly.
> >
> > > "The latent particles $\textbf{z}$ do not correspond to single graphs but encode a *distribution over graphs* $p(G|\textbf{z})$ [...]"
> >
> > Thank you for clarifying this point, and for the additional comment added to the Appendix regarding the difference in annealing compared to the DiBS paper. I appreciate the authors’ argument that this results in greater diversity compared to hard graph samples.
> >
> > Given this choice, I have a few further queries/comments (which I think should also be discussed in the Appendix):
> >
> > - If $\alpha$ is not annealed to $\infty$, then according to Eqn (6) in the DiBS paper, it seems that the distribution $p(\textbf{G}|\textbf{Z})$ will have support over both cyclic and acyclic graphs. If some of the sampled graphs are cyclic, how does your method handle this?
> > - Relating to this, in this case do you anneal the acyclicity parameter $\beta$ to $\infty$?
> > - The computational complexity of the inference phase written down in the new Appendix D.11 is linear in $M$ and $K$. Similarly, the experiment design phase is quadratic in both $M$ and $K$ (if we take the same value of K for the inner and outer expectation). Given this, what is the justification for say taking $M=5$ and $K=40$, rather than say $M=200$ and $K=1$? The latter seems more aligned in spirit with nested estimators.
> > - On a related note, DiBS seems to require quadratic complexity in the number of samples when using SVGD; could you explain why the complexity of the inference phase is linear in $M$?
> >
> > > “Our main conceptual contribution is the integration of causal discovery and reasoning in a Bayesian end-to-end framework [...] “
> >
> > Thank you; I appreciate the conceptual contribution of ABCI as an active learning framework for targeting causal effect queries, which has significant potential and could spur further research in the community. There are still some concerns over the computational cost of the GP-DiBS-ABCI implementation (with regards to the fresh GP inference at each step, and the limitation of interventions being on single nodes), but the detailed discussion regarding computational costs in the response to other reviewers (and the update to the paper), together with the availability of the implementation, make clearer where the bottlenecks are.

---

> > > ### Author Response · Authors · 2022-08-09
> > > **Response to additional questions**
> > >
> > > We thank the reviewer for engaging with our work and for increasing their score.
> > >
> > > We answer their additional questions below.
> > >
> > > > If $\alpha$ is not annealed to $\infty$, then according to Eqn (6) in the DiBS paper, it seems that the distribution will have support over both cyclic and acyclic graphs. If some of the  sampled graphs are cyclic, how does your method handle this?
> > > >
> > > > Relating to this, in this case do you anneal the $\beta$ acyclicity parameter to $\infty$?
> > >
> > >
> > > In our implementation, we do increase the $\beta$ parameter during the SVGD phase to enforce acylicity.
> > > As $\beta\to\infty$, the support of the particle prior $p_\beta(\mathbf{Z})$ reduces to particles $\mathbf{z}$ for which  $p_\alpha(G | \mathbf{z})$ has support only over acyclic graphs.
> > > While it is correct that, for finite $\alpha$, cyclic graphs (in fact, all graphs) have non-zero probability in $p_\alpha(G | \mathbf{z})$, increasing $\alpha\to\infty$ does not enforce acyclicity but forces the mass of $p_\alpha(G | \mathbf{z})$ onto a single graph.
> > > Empirically, we found that $\beta \rightarrow \infty$ suffices for ensuring acyclicity.
> > > In the exceptional case of sampling a cyclic graph during the experimental design phase, we simply discard it.
> > >
> > > > The computational complexity of the inference phase written down in the new Appendix D.11 is linear in $M$ and $K$. Similarly, the experiment design phase is quadratic in both $M$ and $K$ (if we take the same value of $K$ for the inner and outer expectation). Given this, what is the justification for say taking $M=5$ and $K=40$, rather than say $M=200$ and $K=1$? The latter seems more aligned in spirit with nested estimators.
> > >
> > > The reason for using $M=5$ latent particles is computational efficiency. For a fixed budget of, e.g., 200 graph samples, taking $M=200$ and $K=1$ in the *experimental design (ED) phase* may yield better utility estimates.
> > > However, during the DiBS *inference phase*, $M=200$ particles will demand vastly more computation than using only $M=5$ particles (see also our answer below). Hence, by using $M=5$ and $K=40$, we can still obtain a total of 200 graph samples in the ED phase and save on computation during the inference phase; we will add a more detailed account to the next revision.
> > >
> > > We did, however, not conduct a systematic investigation on the optimal choice of $M$ and $K$, as our empirical choices yield good results. Nevertheless, we agree that this is an interesting and relevant question. As we remark in Appendix D.11, we believe that much can be gained in terms of scalability as well as performance by incorporating recent advances in nested Monte-Carlo/information gain estimation techniques in the ED phase.
> > >
> > > > On a related note, DiBS seems to require quadratic complexity in the number of samples when using SVGD; could you explain why the complexity of the inference phase is linear in $M$ ?
> > >
> > > Indeed, we missed an additional term in our analysis; the inference complexity should read $O(T_{SVGD} \cdot (T_{HP} \cdot M \cdot K \cdot d \cdot N^3 + M^2 \cdot d^2))$.
> > > Thank you very much for catching this and pointing it out. We adapted the relevant parts in the Appendix.
> > >
> > > Specifically, a single SVGD step requires to (i) estimate $\nabla_{\mathbf{z}} \log p(\mathbf{z}|\mathbf{x}^{1:t})$ for each z-particle, (ii) compute the particle kernel values (see Eqn (15) in the DiBS paper), and (iii) combine (i) and (ii) to obtain the final particle gradients (cf. Algorithm 1 in the DiBS paper). As described in Appendix D.11, the complexity of (i) is in $O(T_{HP} \cdot M \cdot K \cdot d \cdot N^3)$, whereas the complexity of (ii) is in $O(M^2 \cdot d^2)$ when choosing the latent dimension to equal $d$, and (iii) is in $O(M^2)$. Since (ii) dominates (iii) we need to only include (ii) in the overall complexity of the inference phase.

---

### Official Review · Reviewer_JJRE · 2022-07-12

**Rating:** 7
**Confidence:** 3
**Soundness:** 3 good
**Presentation:** 3 good
**Contribution:** 3 good

**Summary:**

This paper presents a framework to simultaneously infer a causal graph as well as the specific causal model, including specific relationships between the variables from observational data.
To do so, a bayesian approach for joint inference is outlined in the contribution.
The approach also enables learning from additional, new experimental data - in sense of a standard active learning methodology.
A experimental setup justifies the framework as it can show the success of the algorithm.

**Questions:**

* Personally, I consider the terminology using exogenous and endogenous variables for latent and observed variables (L. rather confusing (Def. 1)  somewhat confusing as my understanding is more aligned with variables determined by the model and variables not determined by the model (see https://en.wikipedia.org/wiki/Exogenous_and_endogenous_variables )



**Limitations:**

Limations, especially regarding feasability and tractabilty, are well discussed.

**Strengths And Weaknesses:**



## Strengths

* The problem is well structured, with precise mathematical notation to distinguish certain problem classes

* The choice of the problem - i.e. to extend inferring from only observational to observational and experimental data - is somewhat novel in a sense, that most research in the field is focussed on inferring from observational data alone

* Recent advances, such as the DiBS approach, are considered and brought to use in context of the framework.

* An accompanying codebase provides a computational implementation of the approach, including jupyter notebooks which facilitate understanding the "how-to" of the codebase.
    the problem is well formulated and the manuscript is clear and well written

    the formal style with highlighted Theorems as well as the choice of notation is consistent with the relevant literature of the field

    The real world datasets are well chosen and fit the problem, for the experiments several methods for comparison are included

## Weaknesses

* As the approach outlined is somewhat experimental - in that a "real" active learning approach would necissate manipulation / intervention into the setting - I would have expected a stronger / more elaborate discussion setting,


*   I would have expected to have more discussion of state-of-the-art / comparable methods. The methods for comparison introduced only later could be already discussed theoeretically in an earlier section

* The idea of causal reasoning as formulated in the paper (i.e. via do-formalism interventions) would benefit from some more explanation, ideally using examples.

---

> ### Author Response · Authors · 2022-08-02
> **Response to Reviewer JJRE**
>
> > Personally, I consider the terminology using exogenous and endogenous variables for latent and observed variables [...] somewhat confusing as my understanding is more aligned with variables determined by the model and variables not determined by the model (see [Wikipedia](https://en.wikipedia.org/wiki/Exogenous_and_endogenous_variables) )
>
> The use of "exogenous" and "endogenous" is standard in the causality literature, especially in the context of SCMs:
> "[...] U is a set of background variables, (also called exogenous), that are determined by factors outside the model [...] ", and "[...]  V is a set [...]  of  variables, called endogenous, that are determined by variables in the model -- that is, variables in $U \cup V$" [S1, Def. 7.1.1] (note that in our notation $V$ corresponds to $X$).
> We mostly wanted to emphasise that, as is standard, we assume the exogenous variables to be unobserved (latent) and the endogenous variables to be observed, but did not mean to imply that the terms refer to identical concepts, as you correctly noted. We have adjusted Defn. 1 to reflect this.
>
>
> [S1] Pearl, J. *Causality*. (Cambridge University Press, 2009).

---

### Author Response · Authors · 2022-08-02
**General response to all reviewers and the AC**

We sincerely thank all reviewers for their time and thoughtful feedback. In particular, we are grateful for the largely positive reception of our work:

> "the approach is very appealing because it is highly principled" (`LPpv`)

> "the prose is generally of very high-quality; I found the paper to be very pleasant to read. The description of Bayesian causal inference is thorough, perhaps the clearest that I have seen in the literature." (`8rNg`)

The reviewers seem to agree that our work tackles a "practical and important question" (`T1JL`, `8rNg`), and that it is "precise" (`JJRE`), "solid" (`8rNg`), and "well-written" (`JJRE`, `T1JL`), rating its soundness as *good* (3/4; `JJRE`, `8rNg`, `T1JL`) or *excellent* (4/4; `LPpv`), avg: 3.33; and its presentation as *good* (3/4; `JJRE`, `8rNg`) or *excellent* (4/4; `T1JL`, `LPpv`); avg: 3.5.
We are also glad that `JJRE` appreciated our "jupyter notebooks which facilitate understanding the how-to of the codebase".


The reviewers raised individual points of criticism, which we address in detail in our direct responses to each reviewer. The perhaps main objection seems to stem from `8rNg` who is not yet "entirely convinced that the novelty of the framework itself, or the analysis and insights thereof, is sufficiently strong", rating our contribution as *fair* (2/4).
The other three reviewers, on the other hand, are in agreement that the ABCI approach is “novel” (`T1JL`, `JJRE`) and “original in addressing together the causal discovery and the causal inference questions” (`LPpv`), rating our contribution as *good* (3/4; `T1JL`, `JJRE`) or *excellent* (4/4; `LPpv`).
Indeed, we consider our integrated ABCI framework for learning about an arbitrary causal query $q(\mathcal{M})$ by targeting it with active interventions---instead of a more standard two-stage approach---our main *conceptual* contribution. This complements our concrete GP-DiBS-ABCI implementation, which `8rNg` already described as
> "a sensible and well-engineered end-to-end solution for inferring a target causal query [that] fills a practical gap that previous works did not fully address.”

Following the reviewers' suggestions, we also report the following additional (experimental) results (new or modified material is highlighted in colour in the revised manuscript):

- As suggested by reviewer `T1JL`, *we compare the uncertainties in the causal query vs. the causal graph/model for different utilities*, see Figs. 9 and 10 in Appendix E. The experiments support our theoretical expectations: our utilities yield experiments that are maximally informative about our target query whereas uncertainty about other aspects of the underlying SCM is not considered/marginalised out in this process.
- Prompted by reviewers `LPpv` and `T1JL`, *we discuss the computational complexity of our approach* in a new Appendix D.11. In summary, experimental design and Gaussian Process inference dominate the computational complexity.

We believe that the remarks of the reviewers and the additional results further improved the paper, and will happily respond to any additional questions or comments during the discussion phase.

---

### Meta-Review · Area_Chair_YTud · 2022-08-21

**Recommendation:** Accept
**Confidence:** Certain

**Metareview:**

This paper proposes a Bayesian active learning framework for integrated causal discovery and reasoning. In the framework, one sequentially designs experiments that are maximally informative about a target causal query, collect the corresponding interventional data, update the  beliefs, and repeat. Through simulations, the authors have demonstrated that the approach is more data-efficient than existing methods that only focus on learning the full causal graph. This allows one to accurately learn downstream causal queries from fewer samples.  All the reviews are positive.

**Award:**

No

---

### Decision · Program_Chairs · 2022-09-14

Accept